# LagrangeBench: A Lagrangian Fluid Mechanics Benchmarking Suite

**Artur P. Toshev**[*,1]
artur.toshev@tum.de

**Gianluca Galletti**[*,1]
g.galletti@tum.de

**Fabian Fritz**[1]          **Stefan Adami**[2]          **Nikolaus A. Adams**[1,2]

[1] Chair of Aerodynamics and Fluid Mechanics
Technical University of Munich
85748 Garching b. München, Germany

[2] Munich Institute of Integrated Materials, Energy and Process Engineering
Technical University of Munich
85748 Garching b. München, Germany

## Abstract

Machine learning has been successfully applied to grid-based PDE modeling in various scientific applications. However, learned PDE solvers based on Lagrangian particle discretizations, which are the preferred approach to problems with free surfaces or complex physics, remain largely unexplored. We present LagrangeBench, the first benchmarking suite for Lagrangian particle problems, focusing on temporal coarse-graining. In particular, our contribution is: (a) seven new fluid mechanics datasets (four in 2D and three in 3D) generated with the Smoothed Particle Hydrodynamics (SPH) method including the Taylor-Green vortex, lid-driven cavity, reverse Poiseuille flow, and dam break, each of which includes different physics like solid wall interactions or free surface, (b) efficient JAX-based API with various recent training strategies and three neighbor search routines, and (c) JAX implementation of established Graph Neural Networks (GNNs) like GNS and SEGNN with baseline results. Finally, to measure the performance of learned surrogates we go beyond established position errors and introduce physical metrics like kinetic energy MSE and Sinkhorn distance for the particle distribution. Our codebase is available under the URL: https://github.com/tumaer/lagrangebench.

## 1 Introduction

Partial differential equations (PDEs) are ubiquitous in engineering and physics as they describe the temporal and spatial evolution of dynamical systems. However, for practical applications, a closed analytical solution often does not exist and thus requires numerical methods like finite elements or finite volumes. We can broadly classify such numerical methods into two distinct families: Eulerian and Lagrangian. In Eulerian schemes (grid-based or mesh-based methods), spatially fixed finite nodes, control volumes, cells, or elements are used to discretize the continuous space. In Lagrangian schemes (mesh-free or particle methods), the discretization is carried out using finite material points, often referred to as particles, which move with the local deformation of the continuum. Eulerian and Lagrangian methods offer unique schemes with different characteristics dependent on the PDEs'

---

[*]equal contribution

37th Conference on Neural Information Processing Systems (NeurIPS 2023) Track on Datasets and Benchmarks.

underlying physics. This work exclusively focuses on the physics of fluid flows, i.e., we study the numerical solution to Navier-Stokes equations (NSEs), but offers the machine learning infrastructure for many more Lagrangian problems.

The smoothed particle hydrodynamics (SPH) method is a family of Lagrangian discretization schemes for PDEs, see, e.g., Monaghan [39, 41] and Price [52] for an introduction to SPH. In SPH, the particles move with the local flow velocity, making it a genuine Lagrangian scheme, while the spatial differential operators are approximated using a smoothing operation over neighboring particles. One of the main advantages of SPH compared to Eulerian discretization techniques is that SPH can handle large topological changes with ease since (a) no connectivity constraints between particles are required, and (b) advection is treated exactly [41]. However, SPH is typically more expensive than, e.g., a finite difference discretization on a comparable resolution due to its algorithmic complexity [16]. Despite being initially devised for applications in astrophysics [23, 35], SPH has gained traction as a versatile framework for simulating complex physics, including (but not limited to) multiphase flows [44, 38, 16, 27, 28], internal flows with complex boundaries [1], free-surface flows [4, 36, 37, 40], fluid-structure interactions [3, 68, 69], and solid mechanics [14, 25].

**Eulerian Machine Learning.** In recent years, the application of machine learning (ML) surrogates for PDE modeling has been a very active area of research with approaches ranging from physics-informed neural networks [54], through operator learning [34, 32, 33], to U-Net-based ones [15]. Very recent efforts towards better comparability of such methods have resulted in benchmarking projects led by academia [60] and industry [26]. However, both of these projects exclusively provide the software infrastructure for training models on problems in the Eulerian description and cannot be trivially extended to particle dynamics in 3D space. We note that convolutional neural networks (CNNs) are well suited for problems in the Eulerian description, whereas graph neural networks (GNNs) are best suited to describe points in continuous space. Given the somewhat shifted uprise of research on GNNs with respect to CNNs, we conjecture that particle-based PDE surrogates are still in their early stages of development.

**Lagrangian Machine Learning.** Generic learned physics simulators for Lagrangian rigid and deformable bodies were developed in a series of works using graph-based representations starting with the Interaction Networks [5] and their hierarchical version [46]. There are numerous successors on this line of research mainly based on the graph networks (GN) formalism [6] which improve the network architecture [55, 56], demonstrate great performance on control tasks [30], or introduce interactions on meshes in addition to the interaction radius-based graph [50, 18]. An alternative approach to modeling fluid interactions builds on the idea of continuous convolutions over the spatial domain [65], which has recently been extended to a more physics-aware design by enforcing conservation of momentum [51]. An orthogonal approach to enforcing physical symmetries (like equivariance with respect to the Euclidean group of translation, rotations, and reflections) requires working in the spherical harmonics basis and has been originally introduced for molecular interactions [62, 7, 12], and recently also to fluid dynamics problems [64]. Last but not least, some of the learned particle-based surrogates take inspiration from classical solvers to achieve equivariance [57] or to improve model performance on engineering systems [31].

In terms of data analysis, the closest to our work are those by Li and Farimani [31] and Toshev et al. [64]. In Li and Farimani [31], a dam break and a water fall datasets are presented, the first of which is similar to our dam break, but without explicit wall particles. The Toshev et al. [64] paper uses a 3D TGV and a 3D RPF dataset of every 10th step of the numerical simulator. The fluid datasets from Sanchez-Gonzalez et al. [56] are similar in terms of their physics to the dam break from Li and Farimani [31], and also do not target temporal coarse-graining. We are the first to propose the challenging task of directly predicting every 100th numerical simulator step, on top of which we also introduce 2D/3D LDC, 2D TGV, and 2D RPF. The datasets we present are of diverse and well-studied engineering fluid mechanics systems, each of which has its unique challenges. To the best of our knowledge, there are still no established Lagrangian dynamics datasets to benchmark the performance of new models and we try to fill this gap.

To address the above shortcomings of existing research, we contribute in the following ways:

- **Seven new fluid mechanics datasets** generated using the SPH method. These include the 2D and 3D Taylor-Green vortex (TGV) [9, 10], 2D and 3D reverse Poiseuille flow (RPF) [17], 2D and 3D lid-driven cavity (LDC) [21], and 2D dam break (DAM) [16]. Each of

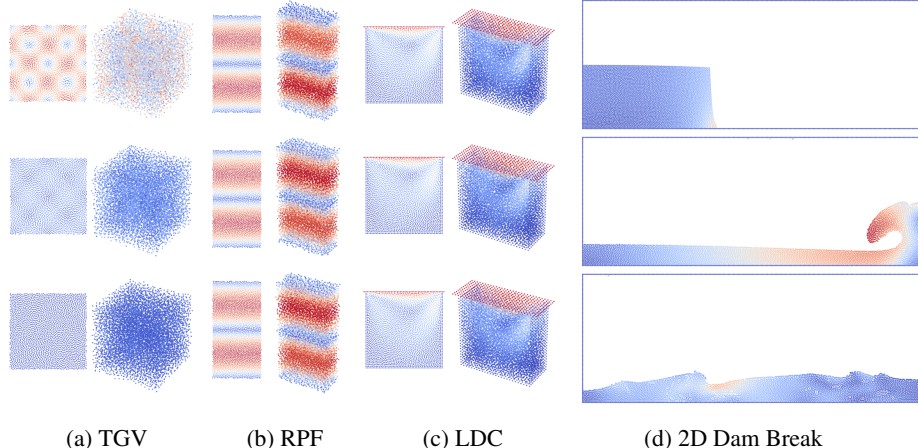

|  (a) TGV | (b) RPF | (c) LDC | (d) 2D Dam Break |

Figure 1: Time snapshots of our datasets, at the initial time (top), 40% (middle), and 95% (bottom) of the trajectory. Color temperature represents velocity magnitude. (a) Taylor Green vortex (2D and 3D), (b) Reverse Poiseuille flow (2D and 3D), (c) Lid-driven cavity (2D and 3D), (d) Dam break (2D).

these datasets captures different dynamics: TGV demonstrates the onset of turbulence, RPF has a spatially dependent external force field, LDC has static and moving wall boundaries, and DAM has a free surface.

- **Efficient JAX-based API** with various recent training strategies including additive random walk-type input noise [56] and the push-forward trick [13]. Also, we offer three neighbor search backends two of which are the JAX-MD implementation [58] and a more memory-efficient version thereof, and the third one being the more flexible CPU-based implementation from `matscipy`.[2]

- **JAX implementation and baseline results of established GNNs** including: GNS [56], EGNN [57], SEGNN [12], and an adapted PaiNN model [59].

## 2   Datasets

To generate the datasets introduced in this work, we use the Lagrangian SPH scheme of Adami et al. [1] to solve the weakly-compressible NSEs [45] in two- and three-dimensional space. A non-dimensional form of the governing equations reads

$$\frac{\mathrm{d}}{\mathrm{d}t}(\rho) = -\rho\left(\nabla \cdot \dot{\mathbf{p}}\right),\tag{1}$$

$$\frac{\mathrm{d}}{\mathrm{d}t}(\dot{\mathbf{p}}) = -\frac{1}{\rho}\nabla p + \frac{1}{\mathrm{Re}}\nabla^2\dot{\mathbf{p}} + \frac{1}{\rho}\mathbf{F},\tag{2}$$

where we denote with $\rho$ the density, $\dot{\mathbf{p}}$ velocity, $p$ pressure, $\mathrm{Re}$ Reynolds number (or the inverse of the nondimensional kinematic viscosity), and $\mathbf{F}$ an external volumetric force field. For weakly-compressible flows, density fluctuations remain small and the pressure $p$ is coupled with the density by the barotropic equation of state $p(\rho) = c_0^2(\rho - \rho_0) + p_{bg}$, where $c_0$ denotes an (artificial) speed-of-sound, $\rho_0$ a reference density, and $p_{bg}$ a background pressure [2].

SPH is a very well-established Lagrangian approximation method of the Navier-Stokes equations. It was introduced in the 70s [23, 35] and has become a standard method for numerical fluid mechanics. The core idea of SPH is to discretize a domain with fluid particles and define the properties of the fluid at some particular locations through a truncated radial kernel interpolation over neighboring particles. By rewriting the NSE in terms of kernel interpolations of the fluid properties (i.e. velocity, pressure, density), we arrive at a system of ODEs for the particle accelerations. By integrating twice we update the velocities and positions of the particles.

---

[2]https://github.com/libAtoms/matscipy

To allow for the most fair runtime comparison in the baselines section, we implemented the SPH method presented in Adami et al. [1] in JAX and plan to open-source our code in the near future. Until then, we include validation results with more details on the solver parameters in Appendix B.

## 2.1 Overview of datasets

Our datasets are generated following a few basic design principles:

1. Each physical system should contain different physics and at the same time have a solid theoretical underpinning in fluid mechanics literature.

2. The Reynolds number is selected as large as possible to still resolve all relevant scales and at the same time to not exceed 10k particles. Above this limit and depending on model choice one would need to discuss domain decomposition and multi-GPU computing, which is beyond the scope of the current work.

3. The number of training instances was deduced on the basis of scaling runs, which we discuss in Section 3. The size of the validation and test runs was chosen to achieve a ratio of training/validation/testing splits of 2/1/1. The reason for the comparably large number of validation and testing instances is that we compute up to 20-step rollout errors which are much noisier than 1-step errors and need to be averaged over more trajectories.

Also, all of our datasets are generated by subsampling every 100th step of the SPH solver to train towards an effective particle acceleration, i.e. temporal coarse-graining. As the stepsize of the numerical solver is governed by the CFL condition (see Equation (19) in Adami et al. [1]), for which we use a speed of sound of $10\times$ the maximum flow velocity and a CFL number of 0.25, then by taking every 100th step the fastest particle would move roughly $2.5\times$ the average particle distance. Learning to directly predict 100 times larger steps is a nontrivial task and given that we use an interaction radius of around $1.5\times$ the average particle distance for the connectivity graph, graph-based approaches need at least two layers to even cover the necessary receptive field.

Table 1: Datasets overview. Either the trajectory length or the number of trajectories is used for splitting into training/validation/testing datasets.

| Dataset | Particle number | Trajectory length | Trajectory count | $\Delta t$ $[\times 10^{-3}]$ | $\Delta x$ $[\times 10^{-3}]$ | Box L×H×D $[-]$ | Re |
|---|---|---|---|---|---|---|---|
| 2D TGV | 2500 | 126 | 100/50/50 | 40 | 20 | 1×1 | 100 |
| 2D RPF | 3200 | 20k/10k/10k | 1 | 40 | 25 | 1×2 | 10 |
| 2D LDC | 2708 | 10k/5k/5k | 1 | 40 | 20 | 1.12×1.12 | 100 |
| 2D DAM | 5740 | 401 | 50/25/25 | 30 | 20 | 5.486×2.12 | 40k |
| 3D TGV | 8000 | 61 | 200/100/100 | 500 | 314.16 | 2π×2π×2π | 50 |
| 3D RPF | 8000 | 10k/5k/5k | 1 | 100 | 50 | 1×2×0.5 | 10 |
| 3D LDC | 8160 | 10k/5k/5k | 1 | 90 | 41.667 | 1.25×1.25×0.5 | 100 |

Table 1 summarizes important physical properties of the datasets including the number of particles and trajectories, as well as the physical time step between training instances $\Delta t$ and the average particle distance $\Delta x$. To generate multiple trajectories of the same system (for TGV and DAM), we randomly draw the positions of the particles and let the system relax under periodic (for TGV) and non-periodic (for DAM) boundary conditions via 1000 steps of SPH relaxation. This relaxed state is then used as the initial state of the trajectories. For the statistically stationary cases (RPF and LDC) the simulation starts with fluid velocities equal to zero and runs until equilibrium. We start collecting data for the datasets after this point. In these cases, the training/validation/testing data is obtained by splitting one very long trajectory in its first half for training, third quarter for validation, and last quarter for testing. All trajectories are generated using the same Reynolds number $Re$, which is also included in Table 1.

Both lid-driven cavity and dam break have solid wall boundaries, implemented using the established generalized wall boundary conditions approach [1]. This approach relies on representing the walls

with "dummy" particles and then 1) enforcing a no-slip boundary condition at the wall surface by assigning to the wall particles the opposite velocity of that of the closest fluid particles, and 2) enforcing impermeability by assigning the same pressure to the wall particles as the pressure of the closest fluid. This approach requires using multiple layers of wall particles (in our case three), but we make the ML task even more difficult by leaving only the innermost wall layer in the dataset.

In the following, we briefly describe the datasets. For more details on the datasets we refer to Appendix C, and for dataset visualizations beyond Figure 1 to Appendix D.

**Decaying Taylor-Green Vortex (TGV).** This problem is characterized by periodic boundary conditions on all sides and a distinct initial velocity field without external driving force, which leads to a flow with decaying kinetic energy under the influence of viscous interactions. This system was first introduced by Taylor and Green [61] in 3D to study the transition of a laminar flow to turbulence. In **two dimensions**, this problem has an analytical solution being the exponentially decaying velocity magnitude and kinetic energy [2]. We initialize the field with the velocity profile from [2] given in Equation (3) with $k = 2\pi$. In **three dimensions**, the case has been explored in literature in-depth, and high-fidelity reference solutions have been generated [9, 10]. Unfortunately, with our limit on particle size of around 10k particles, we cannot resolve the Reynolds numbers from the literature and we chose to work with Re=50, for which we generated our own reference solution using the JAX-Fluids solver [8]. The velocity field for the 3D case is given by Equation (4) with $k = 1$.

$$2D: \quad u = -\cos(kx)\sin(ky), \qquad v = \sin(kx)\cos(ky). \tag{3}$$
$$3D: \quad u = \sin(kx)\cos(ky)\cos(kz), \quad v = -\cos(kx)\sin(ky)\cos(kz), \quad w = 0. \tag{4}$$

**Reverse Poiseuille flow (RPF).** We chose the reverse Poiseuille flow problem [17] over the more popular Poiseuille flow, i.e. laminar channel flow, because it (a) does not require the treatment of boundary conditions as it is fully periodic and (b) has a spatially varying force field. And yet in the laminar case, we still should get the same solution as for the Poiseuille flow. However, our dataset is not fully laminar at $Re = 10$ and some mixing can be observed, making the dynamics more diverse. The physical setup consists of a force field of magnitude 1 in the lower half of the domain and -1 in the upper half. By this constant force, the system reaches a statistically stationary state, which in contrast to the regular Poiseuille flow exhibits rotational motion in the shearing layers between the two opposed streams. The only difference between the **two-dimensional** and **three-dimensional** versions of this dataset is the addition of a third periodic dimension without any significant change in the dynamics.

**Lid-driven cavity (LDC).** This case introduces no-slip boundaries, including a moving no-slip boundary being the lid that drives the flow. Treating wall boundaries with learned methods is a non-trivial topic that is typically treated by either 1) adding node features corresponding to boundary distance [56] or 2) having explicit particles representing the wall [29]. The second approach has a greater generalization capability and we choose to include boundary particles in our datasets. However, instead of keeping all three layers of boundary particles required by the generalized boundary approach [1] in conjunction with the used Quintic spline kernel [45], we just keep the innermost layer by which we significantly reduce the number of particles in the system. The nominally **three-dimensional** version of the LDC problem with periodic boundary conditions in $z$-direction recovers the **two-dimensional** solution and learned surrogates should be able to learn the right dynamics in both formulations.

**Dam break (DAM).** The dam break system [16] is one of the most prominent examples of SPH because it demonstrates its capability of simulating free surfaces with a rather small amount of discretization points. In terms of classical SPH, one major difference between dam break and the previous systems is that the density cannot be simply computed by density summation due to the not full support of the smoothing kernel enforcing the use of density evolution [1]. This is a potential challenge for learned surrogates, on top of which this case is typically simulated as inviscid, i.e. $Re = \infty$. To stabilize the simulation we used the established "artificial viscosity" approach [42] and added on top a very small amount of physical viscosity (leading to $Re = 40k$) to reduce artifacts at the walls. We do not include a 3D version of this dataset as resolving all relevant features would result in exceeding our limit on the number of particles of 10k.

## 2.2 Data format and access, software stack, extensibility

The datasets are stored as HDF5 files with one file for each of the training/validation/testing splits, and one JSON metadata file per dataset. The size of the datasets ranges between 0.3-1.5 GB for the 2D ones and 1.5-2 GB for the 3D ones, resulting in 8 GB total size of all 7 datasets. This size should allow for quick model development and testing, while still covering different types of physics when training on all datasets. The datasets are hosted on Zenodo and can be accessed under the DOI: doi.org/10.5281/zenodo.10021925, [63]. A complete datasheet is provided in Appendix A.

Our benchmarking repository is designed for performance and is built on the JAX library [11] in Python, which is a high-performance numerical computing library designed for machine learning and scientific computing research. We started working on this benchmarking project before the major 2.0 release of the more popular PyTorch library [49], but even after this release introduced many of the speedup benefits of JAX, our experience shows that JAX is faster on graph machine learning tasks at the time of writing this paper. Check Appendix F for detailed speed comparison on a variety of problems. With the simple design of our JAX-based codebase and the use of the established PyTorch `Dataloader`, we plan to maintain the codebase in the foreseeable future. In our codebase we demonstrate how to download and use our datasets, but also how to download, preprocess, and use some of the datasets provided with the Sanchez-Gonzalez et al. [56] paper. We provide many examples of how to set up machine learning training and inference runs using `yaml` configuration files and plan to extend our codebase with more models and optimization strategies in the future.

Current limitations of our framework include that to use the speedup of JAX the ML models need to be compiled for the largest training/inference instance if system sizes vary. This limitation to static shapes for the just-in-time compilation is a known weakness of JAX and we note that it might be an issue to run inference on very large systems. To even allow for a varying number of particles during training we integrated the `matscipy` backend for the neighbors search because, to the best of our knowledge, there is no trivial way to achieve this functionality with the JAX-MD implementation [58]. In addition, we also improve the memory requirements of the JAX-MD neighbor search routine (at the cost of runtime) by serializing a memory-heavy vectorized distance computation over adjacent cells in the cell list. This serialization significantly reduces the memory requirements of the code and could allow for computations with larger batch sizes or larger systems, see Appendix E.

Regarding future extensions of our datasets, we plan to add 1) a multi-phase problem, e.g. Rayleigh-Taylor Instability, and 2) a multi-phase flow with surface tension, e.g. drop deformation in shear flow. By multi-phase, we refer to fluids with different material properties. Introducing such complexity will require learning rich particle embeddings. Surface tension is a phenomenon that relates to how two different media interact with each other, and this would force the learned surrogate to learn complex interactions at low-dimensional manifolds.

## 3 Baselines

### 3.1 Learning problem

We define the task as the autoregressive prediction of the next state of a Lagrangian flow field. For simplicity, we adapt the notation from [56]. Given the state $\mathbf{X}^t$ of a particle system at time $t$, one full trajectory of $K + 1$ steps can be written as $\mathbf{X}^{t_0:t_K} = (\mathbf{X}^{t_0}, \dots, \mathbf{X}^{t_K})$. Each state $\mathbf{X}^t$ is made up of $N$ particles, namely $\mathbf{X}^t = (\mathbf{x}_1^t, \mathbf{x}_2^t, \dots \mathbf{x}_N^t)$, where each $\mathbf{x}_i$ is the state vector of the $i$-th particle. Similarly, the system particle positions at time $t$ are defined as $\mathbf{P}^t = (\mathbf{p}_1^t, \mathbf{p}_2^t, \dots \mathbf{p}_N^t)$. The input node-wise features $\mathbf{x}_i^t$ of particle $i$ are extracted for every time step $t$ based on a window of previous positions $\mathbf{p}_i^{t-H-1:t}$ and optionally other features. In particular, the node features can be selected from:

1. The current position $\mathbf{p}_i^t$.
2. A time sequence of $H$ previous velocity vectors $\dot{\mathbf{p}}_i^{t-H-1:t}$
3. External force vector $\mathbf{F}_i^t$ (if available).
4. The distance to the bounds $\mathbf{B}_i^t$ (optional).

The nodes are connected based on an interaction radius of $\sim 1.5$ times the average interparticle distance similar to what is done in [56], which results in around 10-20 one-hop neighbors. Regarding

the edge features, they can be for example the displacement vectors $\mathbf{d}_{ij}^t$ and/or distances $d_{ij}^t$. The availability of some features is dataset-dependent. For example, the external force vectors are only relevant for reverse Poiseuille and dam break.

For every time step $t$, the model will autoregressively predict the next step particle positions $\mathbf{P}^{(t+1)}$ from the system state $\mathbf{X}^t$ by indirectly estimating acceleration $\ddot{\mathbf{P}}$ or velocity $\dot{\mathbf{P}}$, or by directly inferring the next position $\mathbf{P}$. When using indirect predictions, the next positions $\mathbf{P}^{(t+1)}$ are computed by semi-implicit Euler integration from acceleration or forward Euler from velocity. Finally, the next nodes' state $\mathbf{X}^{(t+1)}$ is computed through feature extraction from the updated positions $\mathbf{P}^{t-H:t+1}$.

## 3.2 Baseline models

We provide a diverse set of graph neural networks [24, 22, 6] adapted for our learning task: GNS [56], which is a popular model for engineering particle problems; EGNN [57], which has shown promising results on N-body dynamics; SEGNN [12], which performs well on similar physical problems [64]; PaiNN [59], which is another simple yet powerful equivariant model from the molecular property-predictions literature. Our choice of GNNs is motivated by their natural extendibility to point cloud data, with relational information between nodes and local neighborhood interactions. Other models not strictly based on Message Passing, such as PointNet++ [53] and Graph UNets [19], have also been considered, but go beyond the scope of the current work. Another non-MP alternative is Graph Transformers [67], but we found them not well-suited for our problems as they would not learn local interactions, which are essential for scaling to larger fluid systems.

**Graph Network-based Simulator (GNS).** The GNS model [56] is a popular learned surrogate for physical particle-based simulations. It is based on the encoder-processor-decoder architecture [6], where the processor consists of multiple graph network blocks [5]. Even though the architecture is fairly simple, as it employs only fully connected layers and layer norm building blocks, GNS has been proven to be capable of modeling both 2D and 3D particle systems. However, performance on long rollouts is unstable and strongly depends on the choice of Gaussian noise to perturb the inputs.

**E(n)-equivariant Graph Neural Network (EGNN).** E(n)-equivariant GNNs are a class of GNNs equivariant with respect to isometries of the $n$-dimensional Euclidean space, namely rotations, translations, and reflections. While other popular equivariant models such as Tensor Field Networks [62], NequIP [7], and SEGNN [12] rely on expensive Clebsch-Gordan tensor product operations to compute equivariant features, the EGNN model [57] is an instance of E($n$) equivariant networks that do not require complex operations to achieve higher order interactions; instead, it treats scalar and vector features separately: the scalars are updated similarly to a traditional GNN, while the vectors (positions) are updated following a layer-stepping method akin to a numerical integrator.

**Steerable E(3)-equivariant Graph Neural Network (SEGNN).** SEGNN [12] is a general implementation of an E(3) equivariant graph neural network. SEGNN layers are directly conditioned on *steerable attributes* for both nodes and edges. The main building block is the steerable MLP: a stack of learnable linear Clebsch-Gordan tensor products interleaved with gated non-linearities [66]. SEGNN layers are message passing layers [22] where Steerable MLPs replace the traditional non-equivariant MLPs for both message and node update functions. The flexibility of the SEGNN framework makes the model well suited for a wide array of physical problems, where vectorial and/or tensorial features can be modeled E(3)-equivariantly. Additionally, the steerable node and edge attributes provide a large flexibility for feature engineering choices. SEGNN is designed to work with 3D data and to make it work on the 2D datasets we set the third dimension to zero; this probably results in a less efficient code, but still preserves equivariance.

**Polarizable atom interaction Neural Network (PaiNN).** PaiNN [59] is yet another variant of E(3) equivariant GNN. Similarly to EGNN, PaiNN models equivariant interactions directly in Cartesian space, meaning separately for scalars and vectors, and does not require tensor products with Clebsch-Gordan coefficients. Because PaiNN was originally designed for molecular property prediction and wasn't intended for general-purpose applications, we had to extend its functionalities in two straightforward yet vital ways: (a) In the original implementation the input vectors are set to the zero vector $\mathbf{0} \in \mathbb{R}^{F \times 3}$ (where $F$ is the latent representation dimension) since there is no directional node information available initially for the problems considered in the paper. We add the possibility of

passing non-zero input vectors, which are lifted to $F$-dimensional tensors by an embedding layer. (b) We include the Gated readout block [59] to predict vectorial features, e.g. acceleration.

**Model naming.** The models are named using the scheme *"{model}-{MP layers}-{latent size}"*. SEGNNs have one additional tag, the superscript "$L = l$", where $l$ is the maximum tensor product order considered in the steerable MLP layers.

### 3.3 Error measures

We measure the performance of the models in three aspects when evaluating on the test datasets:

1. **Mean-squared error** (MSE) of particle positions. $MSE_n$ is the $n$-step average rollout loss.

2. **Sinkhorn distance** as an optimal transport distance measure between particle distributions. Lower values indicate that the particle distribution is closer to the reference one.

3. **Kinetic energy error** $E_{kin}$ ($= 0.5 \sum_i m_i v_i^2$) as a global measure of physical behavior. Especially relevant for statistically stationary datasets (e.g. RPF, LDC) and decaying flows (e.g. TGV). $E_{kin}$ is a scalar value describing all particles at a given time instance, and the $MSE_{E_{kin}}$ is the MSE between the rollout and dataset $E_{kin}$ evolution.

### 3.4 Data scaling

Before we discuss the performance of different GNN baselines, we investigate the complexity of our datasets. One simple way to achieve that is by training one and the same model on different portions of each of the datasets. We chose to train on 1%, 5%, 10%, 20%, 50%, and 100% of our training datasets, and expect to see a saturation of model performance if more data does not add more information to the GNN training. We chose to work with the GNS model as it is probably the fastest (given its simplicity) and most popular one. The number of layers was set to 10 and the latent dimension to 64, which in our experience gives reasonable results. The results are summarized in Figure 2.

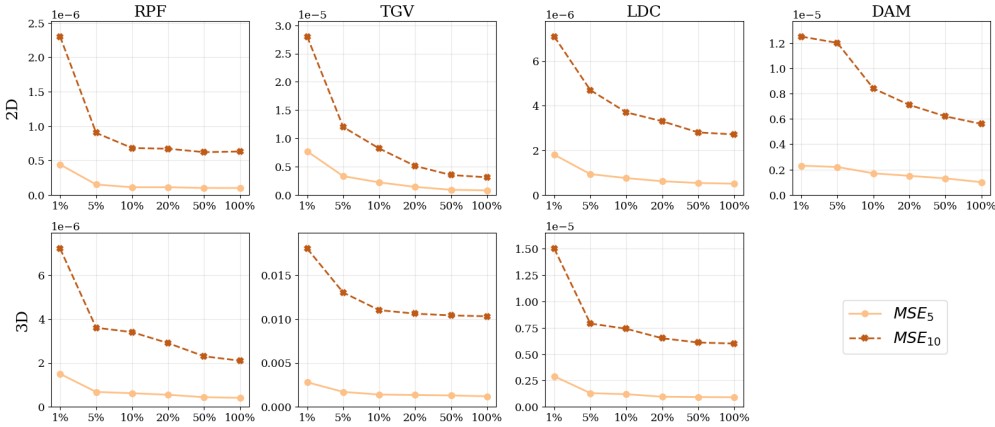

Figure 2: Scaling evaluation on all datasets. The $x$-axis shows the amount of available data, and the $y$-axis shows the position MSE loss values. The model is GNS-10-64 trained for 1M steps (with 40k steps early stopping). Every mark represents a new GNS instance trained with a different amount of data.

We started exploring data scaling with the 2D RPF dataset and that is why it has more training instances than most other datasets (2D RPF and 2D DAM have 20k training frames, whereas all others have 10k). It is 2D RPF from which we saw that 10k samples are enough and that is why most other datasets have roughly 10k training samples. Only dam break seems to keep improving with more data, probably due to the free surface which demonstrates different wave structures, but we decided to restrict the dataset size to twice as much as the others.

## 3.5 Baseline performance

We benchmark each dataset by training every model with different hyperparameters. We observed that higher tensor product orders in SEGNN lead to better performance, but we restrict ourselves to $L = 1$ as higher values are increasingly expensive (see Table 3 for runtimes of $L = \{1, 2\}$). Table 2 shows the best model performance we obtained on all the datasets, and thus sets the state-of-the-art on our datasets. We specify the number of layers and hidden dimensions in the same table and refer to Appendix G for a more complete description of the hyperparameters. Also, we refer to Appendix H for a more complete version of Table 2 including all experiments we conducted. Regarding the effect of the training strategies, we performed a comparison study and summarized the results in Appendix I.

An unfortunate result was that we could not make EGNN and our PaiNN version work well on our datasets. EGNNs are unstable in any configurations we tried, and no model converges to reasonable results. It was also reported in other papers [12] that EGNN gradients are often unstable and produce unfavorable results, especially when predicting the force vectors on larger systems. We leave further investigations to future work. To the best of our knowledge, we are the first to extend PaiNN to general vectorial inputs and consequently apply it to engineering data. We aimed to get similar results to SEGNN with $L = 1$, as in this special case both models only make use of $L = \{0, 1\}$ features. However, the loss we got was on average two orders of magnitude worse than with GNS or SEGNN possibly because, even with our changes, PaiNN is still designed for atomistic systems, which are in a multitude of ways different from our problem. We include our JAX implementation of both EGNN and PaiNN anyway, together with GNS and SEGNN, in the LagrangeBench code repository as working templates for future research. In Appendix G we include more details on what we tried with EGNN and PaiNN.

Looking at Table 2, we see that GNS performs best in most cases with boundary conditions (LDC and DAM) and SEGNN in most pure fluid problems. This result is consistent for smaller and bigger models (see Appendix H) and can be interpreted in two ways: 1) equivariance is not the right inductive bias for problems with boundary conditions, or 2) representing boundaries with one layer of dummy particles is not optimal for equivariant models. We leave this investigation to future work.

Table 2: Baseline results of the best performing models. During training, the best model weights are tracked and saved based on the $MSE_{20}$ loss on the validation dataset. Reported metrics come from the best checkpoint and are averaged on the full test dataset, and over 3 different random seeds. The intervals are the standard deviation over the seeds.

| Dataset | Model | $MSE_5$ | $MSE_{20}$ | Sinkhorn | $MSE_{E_{kin}}$ |
|---|---|---|---|---|---|
| 2D TGV | SEGNN-10-64 | 2.4e−7±5.8e−9 | 4.4e−6±5.8e−9 | 2.1e−7±2.8e−8 | 4.5e−7±8.3e−8 |
| 2D RPF | GNS-10-128 | 1.1e−7±1.2e−9 | 3.3e−6±1.2e−9 | 1.4e−7±2.7e−8 | 1.7e−5±4.4e−7 |
| 2D LDC | GNS-10-128 | 6.4e−7±1.4e−8 | 1.4e−5±1.4e−8 | 1.0e−6±1.2e−7 | 3.7e−3±5.2e−3 |
| 2D DAM | GNS-10-128 | 1.3e−6±7.2e−8 | 3.3e−5±7.2e−8 | 1.4e−5±2.0e−6 | 1.3e−4±1.7e−5 |
| 3D TGV | SEGNN-10-64 | 1.7e−4±3.9e−6 | 5.2e−3±3.9e−6 | 6.4e−6±1.5e−6 | 2.7e−2±4.4e−3 |
| 3D RPF | SEGNN-10-64 | 3.0e−7±2.9e−9 | 1.8e−5±2.9e−9 | 2.9e−7±1.6e−8 | 3.5e−6±1.5e−6 |
| 3D LDC | GNS-10-128 | 7.4e−7±2.1e−8 | 4.0e−5±2.1e−8 | 6.0e−7±1.7e−7 | 2.6e−8±5.0e−9 |

Further, we compare the performance of some instances of our baseline models in terms of inference speed and memory usage in Table 3. It is known that Clebsch-Gordan tensor products are slower than matrix-vector multiplications in fully connected layers, and we see this in the GNS-SEGNN comparison. We didn't expect EGNNs to be noticeably slower than GNS. We noticed in profiling that a large portion of EGNN runtime is spent in computing spatial periodic displacements and shifts (JAX-MD `space` functions, implemented with *modulo* operations). Interestingly, when using non-periodic displacement and shift functions the inference time drops to 3.49ms for the 2D case (3.2K particles) and to 15.1ms for the 3D case (8.1K particles), which is closer to the expected runtime. On the other hand, PaiNN is quite fast to run despite having a significant memory usage likely caused by its design: PaiNN applies a 3× uplifting to the latent embeddings, which are then

Table 3: Inference time and memory usage of the models on 2D and 3D cases (forward only). Runtimes are averaged over 1000 forward passes on one system, RAM is reported as the max memory usage through the rollout. Evaluated on a single Nvidia A6000 48GB GPU.

| Model | Params | Inference [ms] | | Memory [MB] | |
|---|---|---|---|---|---|
| | | 2D (3.2K) | 3D (8.1K) | 2D (3.2K) | 3D (8.1K) |
| GNS-5-64 | 161K | 2.05 | 8.63 | 1121 | 1889 |
| GNS-10-64 | 307K | 3.89 | 16.4 | 1121 | 1889 |
| GNS-10-128 | 1.2M | 6.66 | 32.0 | 1377 | 2913 |
| SEGNN-5-64$^{L=1}$ | 183K | 15.1 | 81.2 | 1377 | 2913 |
| SEGNN-10-64$^{L=1}$ | 360K | 29.7 | 161 | 1379 | 4963 |
| SEGNN-10-64$^{L=2}$ | 397K | 86.0 | 470 | 1893 | 9189 |
| EGNN-5-128 | 663K | 50.0 | 206 | 1377 | 4961 |
| PaiNN-5-128 | 1.0M | 9.09 | 54.2 | 3041 | 17505 |

split into three different representations. This uplifting requires a large matrix multiplication and we suspect that this causes the large memory footprint of PaiNN.

## 4 Discussion

In our work, we propose LagrangeBench, a novel JAX-based framework to develop and evaluate machine learning models on Lagrangian systems. To our knowledge, this is the first formalized benchmarking tool for engineering particle systems, and we consider it a necessary step in the future development of Lagrangian machine learning solvers. We provide a diverse set of 2D and 3D datasets with a solid foundation from engineering fluid simulations. Finally, we include a collection of baseline models originating from multiple machine learning fields and adapt them to our problem setting.

There are many possibilities for extensions of our framework, the most crucial of which is probably implementing a multi-GPU parallelization based on domain decomposition with load balancing. This would allow for training and inference of much bigger systems potentially up to tens of millions of particles with models like Allegro [47, 48]. There are also alternatives to the given `yaml` configuration files in order to automate hyperparameter tuning, which might speed up the development cycles of new machine learning models.

## Acknowledgements

The authors thank Dr. Johannes Brandstetter for his great support in the initial phase of the project including SEGNN reimplementation, push-forward trick, and data analysis. Also, thanks to Christopher Zöller for fruitful discussions on the dataset generation.

## Author Contributions

A.T. and G.G. developed the codebase and ran the experiments together. A.T. led the project and focused on data loading, preprocessing, neighbor search, and reimplemented GNS. G.G. implemented the training pipeline, added the SEGNN, PaiNN, and EGNN models, and engineered the software package. F.F. and S.A. developed the first version of the in-house SPH solver. A.T. extended this solver and generated the datasets advised by F.F. and S.A. in terms of problem selection, literature references, and dataset analysis. N.A. supervised the project from conception to design of experiments and analysis of the results. A.T., G.G., F.F., and N.A. contributed to the manuscript.

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

# A Datasheet

Datasheet adapted from "Datasheets for Datasets" [20]. Our datasets are stored on Zenodo and can be found under the following DOI: 10.5281/zenodo.10021925 [63].

## A.1 Motivation

- ***For what purpose was the dataset created?*** *Was there a specific task in mind? Was there a specific gap that needed to be filled? Please provide a description.*
  The datasets can be used to train autoregressive particle-dynamics surrogates. To the best of our knowledge, these are some of the first Lagrangian fluid dynamics datasets coming from the fluid mechanics literature.

- ***Who created the dataset (e.g., which team, research group) and on behalf of which entity (e.g., company, institution, organization)?***
  The datasets were created by Artur Toshev on behalf of the Chair of Aerodynamics and Fluid Mechanics at the Technical University of Munich.

- ***Who funded the creation of the dataset?*** *If there is an associated grant, please provide the name of the grantor and the grant name and number.*
  No grant.

- ***Any other comments?***
  No.

## A.2 Composition

- ***What do the instances that comprise the dataset represent (e.g., documents, photos, people, countries)?*** *Are there multiple types of instances (e.g., movies, users, and ratings; people and interactions between them; nodes and edges)? Please provide a description.*
  The dataset instances are simulation trajectories of SPH-based fluid system evolutions in time. We generated the datasets by recording every 100th simulation step of the solver to create a challenging temporal coarse-graining benchmark for Lagrangian fluid dynamics.

- ***How many instances are there in total (of each type, if appropriate)?***
  See Table 1 in main paper.

- ***Does the dataset contain all possible instances or is it a sample (not necessarily random) of instances from a larger set?*** *If the dataset is a sample, then what is the larger set? Is the sample representative of the larger set (e.g., geographic coverage)? If so, please describe how this representativeness was validated/verified. If it is not representative of the larger set, please describe why not (e.g., to cover a more diverse range of instances, because instances were withheld or unavailable).*
  We provide all datasets that were used for training our machine-learning models and generating the benchmarking results.

- ***What data does each instance consist of?*** *"Raw" data (e.g., unprocessed text or images) or features? In either case, please provide a description.*
  Each of our 7 datasets consists of 4 main files: "train.h5", "valid.h5", "test.h5", "metadata.json", the first three of which are 2/1/1 splits of the SPH simulation trajectories for training/validation/testing, and the metadata file contains the parameters used to generate the dataset. Each data split is an HDF5 file with the first dictionary keys being the trajectory number as a string ("00000", "00001", ...), and each trajectory containing two single precision arrays named "position" (shape: [# time steps, # particles, dimension]) and "particle_type" (shape: [# particles]).

  In addition, the datasets that have external forces, i.e. RPF and DAM, have a fifth file `force.py`, which defines the force acting on a particle based on its coordinates.

- ***Is there a label or target associated with each instance?*** *If so, please provide a description.*
  Yes. The target in autoregressive dynamics prediction is the next instance in time.

- ***Is any information missing from individual instances?*** *If so, please provide a description, explaining why this information is missing (e.g., because it was unavailable). This does not include intentionally removed information, but might include, e.g., redacted text.*
  No.

- *Are relationships between individual instances made explicit (e.g., users' movie ratings, social network links)? If so, please describe how these relationships are made explicit.*
  N/A.

- *Are there recommended data splits (e.g., training, development/validation, testing)? If so, please provide a description of these splits, explaining the rationale behind them.*
  The provided 2/1/1 split (see the fourth question of this subsection) is designed to provide enough validation and testing instances for long rollout evaluation.

- *Are there any errors, sources of noise, or redundancies in the dataset? If so, please provide a description.*
  The datasets directly represent the dynamics of the described ground-truth SPH solver.

- *Is the dataset self-contained, or does it link to or otherwise rely on external resources (e.g., websites, tweets, other datasets)? If it links to or relies on external resources, a) are there guarantees that they will exist, and remain constant, over time; b) are there official archival versions of the complete dataset (i.e., including the external resources as they existed at the time the dataset was created); c) are there any restrictions (e.g., licenses, fees) associated with any of the external resources that might apply to a dataset consumer? Please provide descriptions of all external resources and any restrictions associated with them, as well as links or other access points, as appropriate.*
  Yes.

- *Does the dataset contain data that might be considered confidential (e.g., data that is protected by legal privilege or by doctor–patient confidentiality, data that includes the content of individuals' non-public communications)? If so, please provide a description.*
  No.

- *Does the dataset contain data that, if viewed directly, might be offensive, insulting, threatening, or might otherwise cause anxiety? If so, please describe why.*
  No.

- *Any other comments?*
  No.

### A.3 Collection Process

- *How was the data associated with each instance acquired? Was the data directly observable (e.g., raw text, movie ratings), reported by subjects (e.g., survey responses), or indirectly inferred/derived from other data (e.g., part-of-speech tags, model-based guesses for age or language)? If the data was reported by subjects or indirectly inferred/derived from other data, was the data validated/verified? If so, please describe how.*
  The data was generated by performing SPH simulations.

- *What mechanisms or procedures were used to collect the data (e.g., hardware apparatus(es) or sensor(s), manual human curation, software program(s), software API(s))? How were these mechanisms or procedures validated?*
  The simulations were performed using our in-house JAX-based (in Python) SPH solver. The postprocessing was done by means of Python scripts.

- *If the dataset is a sample from a larger set, what was the sampling strategy (e.g., deterministic, probabilistic with specific sampling probabilities)?*
  See the third question of the previous section on "Composition".

- *Who was involved in the data collection process (e.g., students, crowdworkers, contractors) and how were they compensated (e.g., how much were crowdworkers paid)?*
  Artur Toshev set up and ran the numerical simulations, and performed the postprocessing. See "Acknowledgements" and "Author Contributions" for further people involved in the dataset design. There were no crowdworkers.

- *Over what timeframe was the data collected? Does this timeframe match the creation timeframe of the data associated with the instances (e.g., recent crawl of old news articles)? If not, please describe the timeframe in which the data associated with the instances was created.*
  The datasets were generated between July 2022 and May 2023.

- *Were any ethical review processes conducted (e.g., by an institutional review board)? If so, please provide a description of these review processes, including the outcomes, as well as a link or other access point to any supporting documentation.*
No.

- *Any other comments?*
No.

## A.4 Preprocessing/cleaning/labeling

- *Was any preprocessing/cleaning/labeling of the data done (e.g., discretization or bucketing, tokenization, part-of-speech tagging, SIFT feature extraction, removal of instances, processing of missing values)? If so, please provide a description. If not, you may skip the remaining questions in this section.*
We subsampled the full solver solution at every 100th step to achieve a temporal coarse-graining problem.

- *Was the "raw" data saved in addition to the preprocessed/cleaned/labeled data (e.g., to support unanticipated future uses)? If so, please provide a link or other access point to the "raw" data.*
We used the full trajectories to validate the SPH solver but never used them to train machine learning models.

- *Is the software that was used to preprocess/clean/label the data available? If so, please provide a link or other access point.*
Not yet.

- *Any other comments?*
No.

## A.5 Uses

- *Has the dataset been used for any tasks already? If so, please provide a description.*
Not beyond this paper.

- *Is there a repository that links to any or all papers or systems that use the dataset? If so, please provide a link or other access point.*
We plan on listing papers that use our datasets in the LagrangeBench Github repository: https://github.com/tumaer/lagrangebench.

- *What (other) tasks could the dataset be used for?*
Any task related to learning the Navier-Stokes dynamics based on particle discretizations.

- *Is there anything about the composition of the dataset or the way it was collected and preprocessed/cleaned/labeled that might impact future uses? For example, is there anything that a dataset consumer might need to know to avoid uses that could result in unfair treatment of individuals or groups (e.g., stereotyping, quality of service issues) or other risks or harms (e.g., legal risks, financial harms)? If so, please provide a description. Is there anything a dataset consumer could do to mitigate these risks or harms?*
No.

- *Are there tasks for which the dataset should not be used? If so, please provide a description.*
No.

- *Any other comments?*
No.

## A.6 Distribution

- *Will the dataset be distributed to third parties outside of the entity (e.g., company, institution, organization) on behalf of which the dataset was created? If so, please provide a description.*
Yes, the dataset is freely and publicly available and accessible.

- *How will the dataset be distributed (e.g., tarball on website, API, GitHub)? Does the dataset have a digital object identifier (DOI)?*

The datasets will be distributed as zip files through Zenodo under the DOI [doi.org/10.5281/zenodo.10021925](doi.org/10.5281/zenodo.10021925).

- ***When will the dataset be distributed?***
  The dataset is on Zenodo as of October 2023.

- ***Will the dataset be distributed under a copyright or other intellectual property (IP) license, and/or under applicable terms of use (ToU)?*** *If so, please describe this license and/or ToU, and provide a link or other access point to, or otherwise reproduce, any relevant licensing terms or ToU, as well as any fees associated with these restrictions.*
  The dataset is licensed under "Creative Commons Attribution 4.0 International".

- ***Have any third parties imposed IP-based or other restrictions on the data associated with the instances?*** *If so, please describe these restrictions, and provide a link or other access point to, or otherwise reproduce, any relevant licensing terms, as well as any fees associated with these restrictions.*
  No.

- ***Do any export controls or other regulatory restrictions apply to the dataset or to individual instances?*** *If so, please describe these restrictions, and provide a link or other access point to, or otherwise reproduce, any supporting documentation.*
  No.

- ***Any other comments?***
  No.

## A.7    Maintenance

- ***Who will be supporting/hosting/maintaining the dataset?***
  The dataset will be hosted on Zenodo.

- ***How can the owner/curator/manager of the dataset be contacted (e.g., email address)?***
  Emails of the corresponding authors are on the first page of the main paper.

- ***Is there an erratum?*** *If so, please provide a link or other access point.*
  No. Zenodo allows versioning.

- ***Will the dataset be updated (e.g., to correct labeling errors, add new instances, delete instances)?*** *If so, please describe how often, by whom, and how updates will be communicated to dataset consumers (e.g., mailing list, GitHub)?*
  If there are changes to the dataset, these will be made and documented on Zenodo.

- ***If the dataset relates to people, are there applicable limits on the retention of the data associated with the instances (e.g., were the individuals in question told that their data would be retained for a fixed period of time and then deleted)?*** *If so, please describe these limits and explain how they will be enforced.*
  N/A.

- ***Will older versions of the dataset continue to be supported/hosted/maintained?*** *If so, please describe how. If not, please describe how its obsolescence will be communicated to dataset consumers.*
  Versioning will be managed by Zenodo.

- ***If others want to extend/augment/build on/contribute to the dataset, is there a mechanism for them to do so?*** *If so, please provide a description. Will these contributions be validated/verified? If so, please describe how. If not, why not? Is there a process for communicating/distributing these contributions to dataset consumers? If so, please provide a description.*
  The issue system on the project Guthub can be used to report bugs and suggest dataset extensions. Using custom datasets on top of the provided ones is already discussed in the paper and our codebase seamlessly allows for it (see "notebooks" in GitHub repository).

- ***Any other comments?***
  No.

# B Solver validation

A crucial part of numerical solver development is the validation. This is the process of reproducing the dynamics of systems either 1) with an analytical solution or 2) with a high-quality reference solution from another numerical simulation. We chose two example systems for validation, namely the Poiseuille flow and the dam break systems.

## B.1 Poiseuille flow

Here, we reproduce the Poiseuille flow case described in Morris et al. [45]. The only technical difference is that we first non-dimensionalize the problem before we run the simulation, but this procedure does not influence the Reynolds number and thus the behavior of the system. For more details on non-dimensionalization, we refer to Bezgin et al. [8]. The reference values we use are the density $\rho_{ref} = 1000\,kg/m^3$, length $L_{ref} = 0.001\,m$, and velocity $V_{ref} = 1e-5\,m/s$. Our dimensionless results almost perfectly match the analytical series expansion solution with 60 particles spanning the width of the channel, see Figure 3.

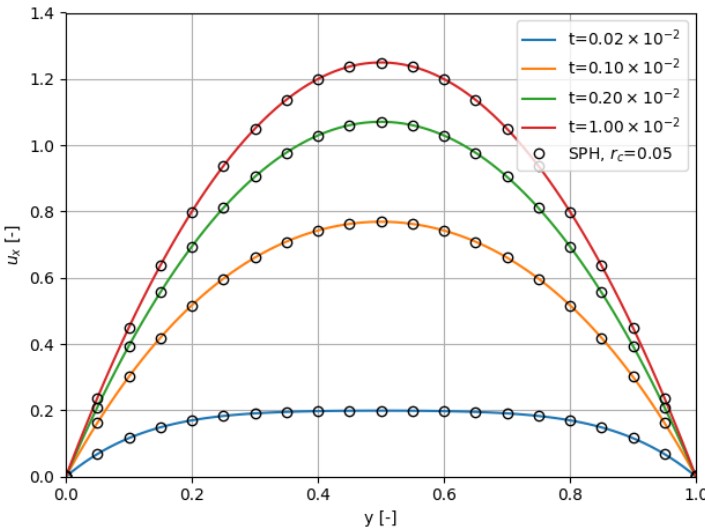

Figure 3: Comparison of SPH ($\circ$) and series solutions ($-$) for Poiseuille flow at $Re = 0.0125$.

## B.2 Dam break

We recover the dam break simulation very closely to the reference result in Adami et al. [1]. Here, we include a quantitative validation of the pressure on the right wall at height $0.2$ (see Figure 4), as well as a qualitative comparison of the pressure distribution at time instances $t = 2, 5.7, 6.2,$ and $7.4$ (see Figure 5).

# C Dataset details

Here we give more details on the dataset-generation process by continuing Table 1 with Table 4. The properties we present are 1) the boundary conditions type of the computational domain, 2) the magnitude of external force $F_{mag}$, 3) the quantities from the equation of state in Section 2 being $c_0$, $\rho_0$, and $p_{bg}$, 4) the length of one trajectory of a given type in dimensionless time units, 5) the viscosity, which is proportional to $Re^{-1}$, and 6) the size of each dataset in GB.

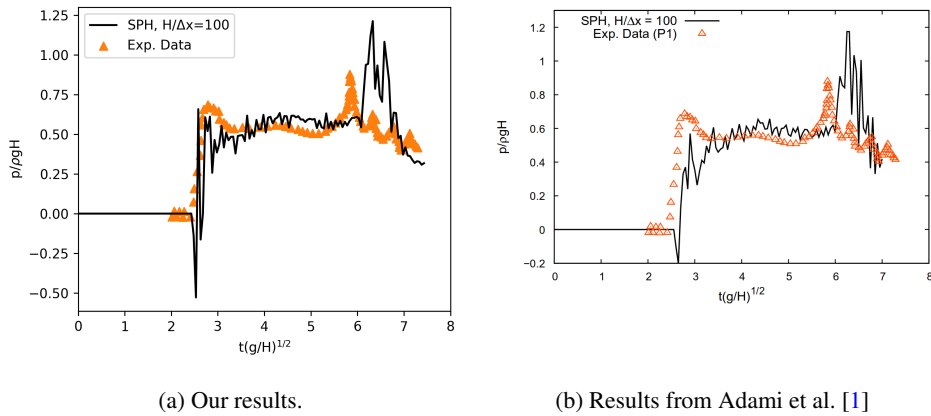

(a) Our results.  (b) Results from Adami et al. [1]

Figure 4: Temporal pressure evolution at $y/H = 0.2$ on the right wall.

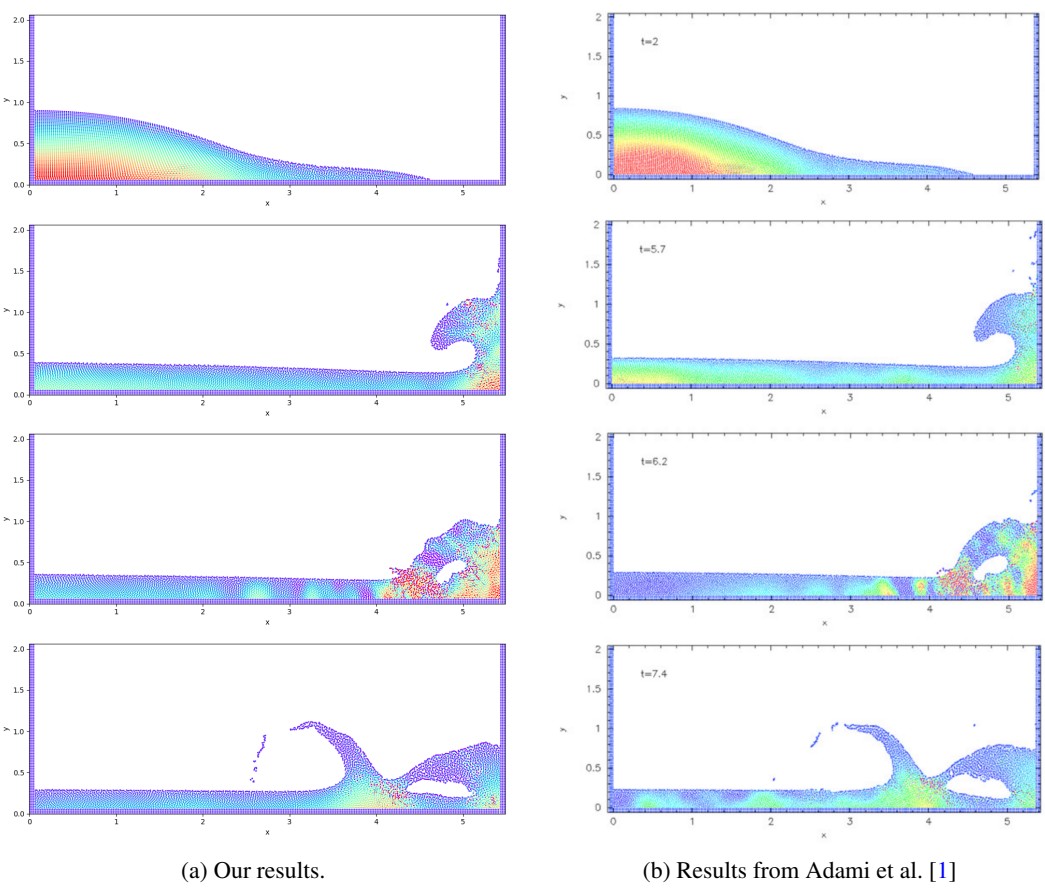

(a) Our results.  (b) Results from Adami et al. [1]

Figure 5: Pressure distribution along dam break simulation at $t = 2, 5.7, 6.2,$ and $7.4$.

## D   Dataset visualizations

Figures 6, 7, 8, 9 and Figures 10, 11, 12 show the time evolution during the rollout of 2D and 3D datasets, espectively. Each dot represents a particle, and a warmer color represents a higher velocity magnitude. In the cases of 6, 7, 8 the (approximately) correct scale is kept.

Table 4: Datasets overview. Either the trajectory length or the number of trajectories is used for splitting into training/validation/testing datasets.

| Dataset | periodic domain | $F_{mag}$ | $c_0$ | $\rho_0$ | $p_{bg}$ | SPH runtime 100 steps [ms] | viscosity | size [GB] |
|---|---|---|---|---|---|---|---|---|
| 2D TGV | yes | 0 | 10 | 1 | 0 | 39.1 | 0.01 | 0.43 |
| 2D RPF | yes | 1 | 10 | 1 | 5 | 43.0 | 0.1 | 0.88 |
| 2D LDC | only lid | 0 | 10 | 1 | 1 | 51.2 | 0.01 | 0.34 |
| 2D DAM | no | 1 | 14.14 | 1 | 0 | 89.4 | 5e$-$5 | 1.4 |
| 3D TGV | yes | 0 | 10 | 1 | 0 | 388 | 0.02 | 2.0 |
| 3D RPF | yes | 1 | 10 | 1 | 2 | 424 | 0.1 | 1.6 |
| 3D LDC | lid and in $z$ | 0 | 10 | 1 | 1 | 678 | 0.01 | 1.4 |

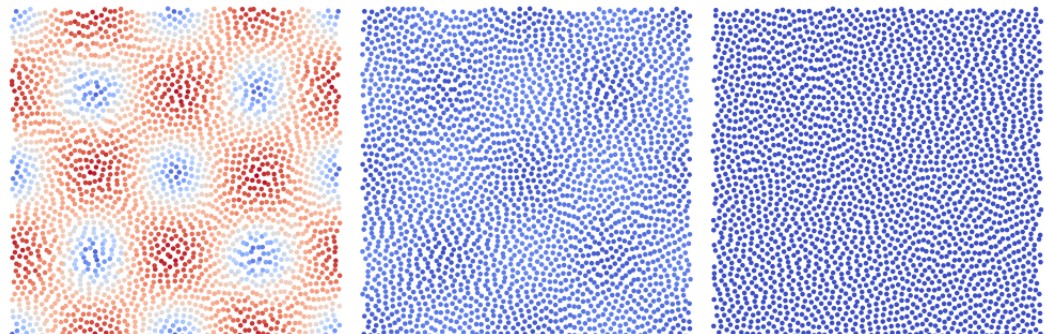

Figure 6: Taylor Green vortex velocity snapshot at $step = 10, 60,$ and $113$.

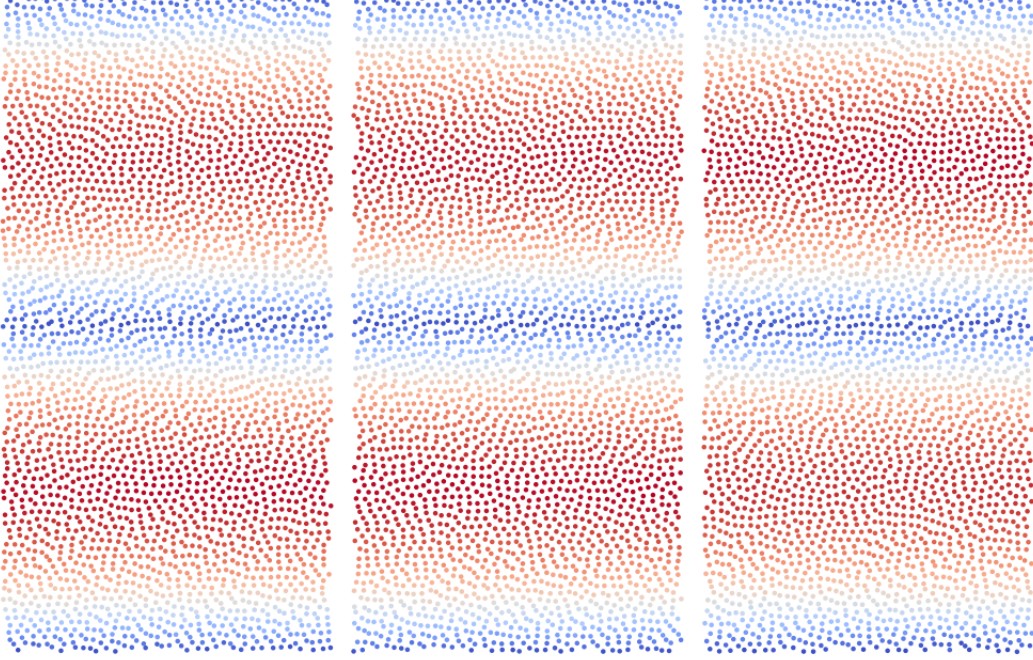

Figure 7: Reverse Poiseuille flow velocity snapshot at $step = 10, 9600,$ and $18000$.

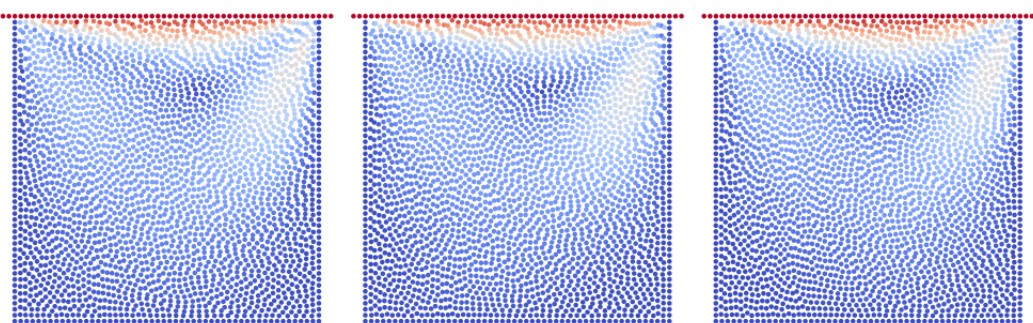

Figure 8: Lid driven cavity velocity snapshot at $step = 10, 4800,$ and $9000$.

Figure 9: Dam break velocity snapshot at $step = 10, 192,$ and $360$.

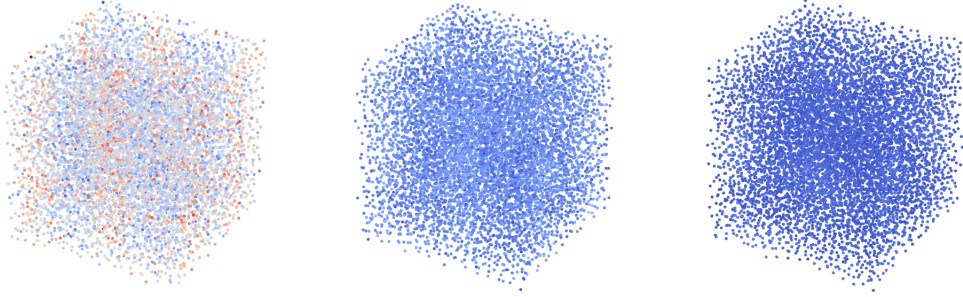

Figure 10: Taylor Green vortex 3D velocity snapshot at $step = 10, 29,$ and $54$.

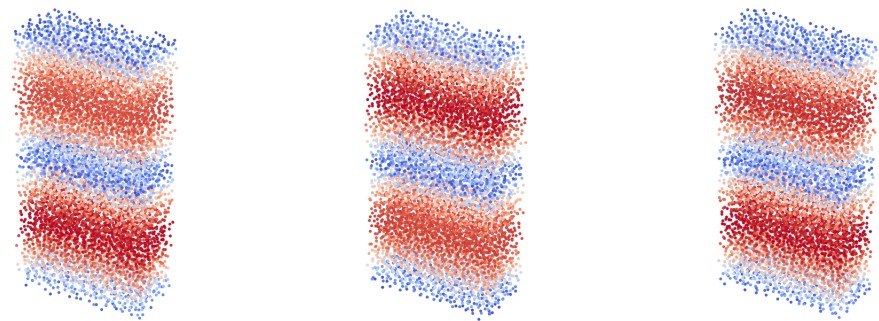

Figure 11: Reverse Poiseuille flow velocity snapshot at $step = 10, 4800,$ and $9000$.

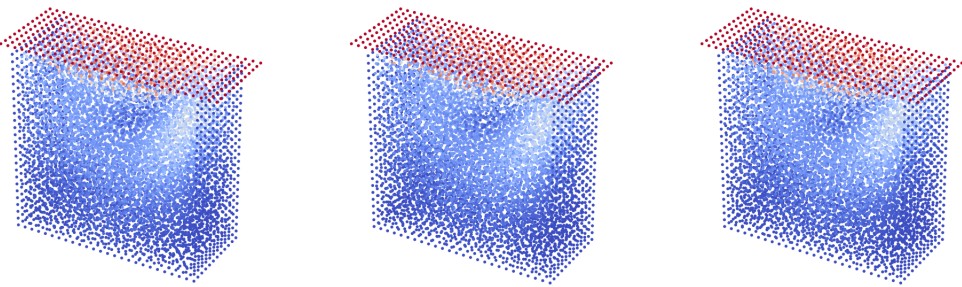

Figure 12: Lid driven cavity 3D velocity snapshot at $step = 10, 4800,$ and $9000$.

# E  Neighbor search implementations

We integrate three neighbor list routines in our codebase: original jax-md implementation (`jaxmd_vmap`), serialized version thereof (`jaxmd_scan`), and a wrapper around the matscipy neighbor list algorithm (`matscipy`).

`jaxmd_vmap` refers to using the original cell list-based implementation from the JAX-MD[3] library. This one can be seamlessly integrated into our JAX-based API and is the preferred choice for systems with a constant number of particles.

`jaxmd_scan` refers to using a more memory-efficient implementation of the JAX-MD function. We achieve this by partitioning the search over potential neighbors from the cell list-based candidate neighbors into $M$ chunks. We need to define three variables to explain how our implementation works:

- $X \in \mathbb{R}^{N \times d}$ - the particle coordinates of $N$ particles in $d$ dimensions.

- $h \in \mathbb{N}^N$ - the list specifying to which cell a particle belongs.

- $L \in \mathbb{N}^{C \times cand}$ - list specifying which particles are potential candidates to a particle in cell $c \in [1, ..., C]$. The number of potential candidates $cand$ is the product of the fixed cell capacity and the number of reachable cells, e.g. 27 in 3D.

The `jaxmd_vmap` implementation essentially instantiates all possible connections by creating an object of size $N \cdot cand$, and only after all distances between potential neighbors have been computed the edge list is pruned to its actual size being roughly $6\times$ smaller in 3D. This factor comes from the fact that the cell size is approximately equal to the cutoff radius and if we split a unit cube into $3^3$ cells, then the volume of a sphere with $r = 1/3$ will be around $1/6$ the volume of the cube. By splitting $X$ and $h$ into $M$ parts and iterating over $L$ with a `jax.lax.scan` loop, we can remove $5/6$ of the edges before putting them together into one list.

`matscipy` is used to enable computations over systems with a variable number of particles, for which none of the above implementations can be used. We essentially wrap the neighbor search routine from `matscipy.neighbours.neighbour_list` [4] with a `jax.pure_callback`. This is again a cell list-based algorithm, however only available on CPU. Our wrapper essentially mimics the behavior of the JAX-MD function, but pads all non-existing particles to the maximal number of particles in the dataset.

## E.1  Performance

To demonstrate the improvement we get from `jaxmd_scan` relative to `jaxmd_vmap`, we compare the largest number of particles whose neighbor list computation fits into memory. We ran the comparison on an A6000 GPU with 48GB memory and observed that the default vectorized implementation can handle up to 1M particles before running out of memory, while our serialized implementation reaches 3.3M. This happens at almost no additional time cost and holds for both allocating a system and updating it after compilation.

The `matscipy` implementation itself is very fast for small systems (10k particles) and doesn't take any GPU memory for the construction of the edge list, but due to copying data between CPU and GPU, we observed a slowdown in training a SEGNN on 2D DAM in the order of $+50\%$. In addition, it seems that `matscipy` uses a single CPU computation and is therefore limited to smaller systems (see this timing comparison https://github.com/arturtoshev/pbc_neighbors_benchmark).

Overall, we observe reasonable performance from each of these implementations on systems with up to 10k particles, but more investigations need to be conducted toward comparing these algorithms on larger systems.

---

[3]https://github.com/jax-md/jax-md
[4]https://github.com/libAtoms/matscipy

# F  JAX vs PyTorch GNN performance comparison

We compare the speed of PyTroch vs JAX on all four models (GNS, SEGNN, EGNN, PaiNN). We note that all models have existing implementations in PyTroch and all were reimplemented by us in JAX. For each model, we consider datasets and experiments from the original paper, namely:

- GNS - we use the `WaterDropSample` dataset from Sanchez-Gonzalez et al. [56] (available under github.com/deepmind/deepmind-research/tree/master/learning_to_simulate), which is a subset of a 2D fluid dataset with up to 1000 particles. As a reference PyTorch implementation, we use github.com/wu375/simple-physics-simulator-pytorch-geometry, and we average the forward evaluation time for one particular system with 803 particles over 5 rollouts of 1000 steps each. The neighbor search with LagrangeBench uses `matscipy`.

- EGNN - we evaluate runtime on the $NBody_{(5)}$ dataset [57], a custom 3D particle dataset with 5 electrical charges in free space. The reference PyTorch implementation is the official EGNN github.com/vgsatorras/egnn. Results are reported on batches of 100 systems (500 particles per batch).

- SEGNN - similarly to EGNN, we measure SEGNN runtime on $NBody_{(5)}$ and $NBody_{(100)}$, which are 5 and 100 particle systems respectively [57, 12]. The reference PyTorch implementation is the official SEGNN github.com/RobDHess/Steerable-E3-GNN. Inference is over a batch of 100 samples on the respective dataset (500 and 10000 particles per batch).

- PaiNN - we evaluate runtime on QM9 [43]. For the Jax implementation, batches must be padded to the worst possible size (because of static shapes). The reference PyTorch implementation as well as QM9 data access is adapted from the official implementation from SchnetPack github.com/atomistic-machine-learning/schnetpack. Results are reported on batches of 100 systems (variable number of nodes).

Table 5: Times [ms] per step for JAX and PyTorch inference.

|       | GNS | SEGNN | | EGNN | PaiNN |
|-------|-----|-------|-------|-------|-------|
|       | WaterDropSample | $NBody_{(5)}$ | $NBody_{(100)}$ | $NBody_{(5)}$ | QM9 |
| Torch | 9.2 | 21.22 | 60.55 | 8.27 | 163.23 |
| Jax   | 8.0 | 3.77  | 41.72 | 0.94 | 8.42 |

# G  Benchmarking hyperparameters

The exact model configurations for SEGNN, GNS, EGNN, and PaiNN can be found in Table 6. In particular, the hyperparameters are split into 4 groups, namely:

1. *Model architecture*. Describe the model structure.
   - *MP layers*. Message passing steps.
   - *Latent size*. Hidden layer dimension.
   - *MLP blocks*. Number of *"linear-activation"* blocks in the update-message functions.
   - $L_{max}$ *hidden*. Cutoff tensor product level in the hidden layers. SEGNN only.
   - $L_{max}$ *attrributes*. Max spherical harmonics level for attributes. SEGNN only.
2. *Optimization*. Learning rate scheduler settings.
   - *Weight decay*. Weight decay regularization.
   - *LR initial*. Starting learning rate.
   - *LR final*. Minimum learning rate.
   - *LR decay steps*. Number of steps to decay from *LR initial* to *LR final*
3. *Training strategies*. Noise and push-forward.
   - *Noise std*. Gaussian random walk noise standard deviation.
   - *PF steps*. Number of push-forward steps.
   - *PF probs*. Probability of sampling each push-forward step number.

4. *Normalization*. Network and input normalization
   - *Normalization*. Network normalization type.
   - *Isotropic inputs*. Whether to normalize all input channels with the same statistics.

Training the baselines takes several days, especially the larger instances of SEGNN. In order to reduce the amount of training runs we prioritized architecture/model-related values over arguably less impactful optimization hyperparameters, such as *LR decay steps* and the *Weight decay*. Increasing or decreasing the number of *MLP blocks* would respectively considerably grow the parameter count or reduce the nonlinearity of the network (since the last update layer is usually linear). We investigated in-depth non-isotropic input normalization and found it advantageous since normalizing the input dimensions independently leads to a more favorable data distribution. However, because independent rescaling of vector dimensions via normalization breaks equivariance, all equivariant models use isotropic normalization by default.

Table 6: Hyperparameter configuration for SEGNN, GNS, EGNN, and PaiNN. Push-forward settings are the same for all models. Noise levels depend on the dataset.

| | SEGNN-5-64 | SEGNN-10-64 | GNS-5-64 | GNS-10-128 | EGNN-5-128 | PaiNN-5-128 |
|---|---|---|---|---|---|---|
| MP layers | 5 | 10 | 5 | 10 | 5 | 5 |
| Latent size | 64 | 64 | 64 | 128 | 128 | 128 |
| MLP blocks | 2 | 2 | 2 | 2 | 1 *or* 2 | 1 *or* 2 |
| $L_{max}$ hidden | 1 | 1 | NA | NA | NA | NA |
| $L_{max}$ attributes | 1 | 1 | NA | NA | NA | NA |
| Params | 183K | 360K | 161K | 1.2M | 663K | 1.0M |
| Weight decay | 1e-8 | 1e-8 | 1e-8 | 1e-8 | 1e-8 | 1e-8 |
| LR initial | 5e-4 | 5e-4 | 5e-4 | 1e-4 | 5e-4 | 5e-4 |
| LR final | 1e-6 | 1e-6 | 1e-6 | 1e-6 | 1e-6 | 1e-6 |
| LR decay steps | 1e5 | 1e5 | 1e5 | 1e5 | 5e4 | 5e4 |
| Noise std | data-dependent. $\in [3e-4, 1e-3]$ | | | | | |
| PF steps | up to 3 | | | | | |
| PF probs | $[0.8, 0.1, 0.05, 0.05]$ | | | | | |
| Normalization | No | No | Layer | Layer | No | No |
| Isotropic inputs | Yes | Yes | No | No | Yes | Yes |

Table 6 only shows a single configuration for EGNN and PaiNN. For EGNN, we found that going beyond 5 message passing steps resulted in unstable gradients. The poor stability of EGNN on larger systems has already been documented in other works [12]. For PaiNN, following the approach of the original paper [59], we preferred a wider model with fewer message passing layers and a large (cutoff) interaction radius.

It is worth noting that Table 6 does not capture the full extent of EGNN and PaiNN experiments. Both models were evaluated and experimented in conceptually different setups, namely:

- EGNN was experimented on 1) original N-body formulation with position updates, and 2) directly working with velocities without updating positions. However, we found that only the first approach behaved remotely reasonably.
- Different PaiNN setups were also evaluated: with/without cutoff functions or radial embeddings, including different features in the nodes, and with/without our general vector input change. We observed that a (cosine) cutoff of 1.5 times the average inter-particle distance, together with a learnable Gaussian RBF behaved the best.

# H  More baseline results

The most relevant results from our experiment runs are in Table 7. Only $MSE_5$ and $MSE_{20}$ metrics are reported for the sake of simplicity and runtime. Wherever a cell is marked with *"unstable"* it means that the model either constantly got large errors or that the metric was not stable during training.

Table 7: Benchmarking results from all experiments. The results included in Table 2 are marked in bold.

| Dataset | Model | $MSE_5$ | $MSE_{20}$ | Inference [ms] |
|---|---|---|---|---|
| TGV 2D | GNS-5-64 | 6.4e−7 | 9.6e−6 | 1.43 |
| | GNS-10-128 | 3.9e−7 | 6.6e−6 | 3.43 |
| | SEGNN-5-64$^{L=1}$ | 3.8e−7 | 6.5e−6 | 9.78 |
| | **SEGNN-10-64$^{\mathbf{L=1}}$** | 2.4e−7 | 4.4e−6 | 20.24 |
| RPF 2D | GNS-5-64 | 4.0e−7 | 9.8e−6 | 2.05 |
| | **GNS-10-128** | 1.1e−7 | 3.3e−6 | 3.89 |
| | SEGNN-5-64$^{L=1}$ | 1.3e−7 | 4.0e−6 | 15.11 |
| | SEGNN-10-64$^{L=1}$ | 1.3e−7 | 4.0e−6 | 29.65 |
| | EGNN-5-128 | *unstable* | *unstable* | 60.77 |
| | PaiNN-5-128 | 3.0e−6 | 7.2e−5 | 9.09 |
| LDC 2D | GNS-5-64 | 2.0e−6 | 1.7e−5 | 1.43 |
| | **GNS-10-128** | 6.4e−7 | 1.4e−5 | 3.43 |
| | SEGNN-5-64$^{L=1}$ | 9.9e−7 | 1.7e−5 | 9.78 |
| | SEGNN-10-64$^{L=1}$ | 1.4e−6 | 2.5e−5 | 20.24 |
| DAM 2D | GNS-5-64 | 2.1e−6 | 6.3e−5 | 3.94 |
| | **GNS-10-128** | 1.3e−6 | 3.3e−5 | 9.68 |
| | SEGNN-5-64$^{L=1}$ | 2.3e−6 | 7.5e−5 | 28.82 |
| | SEGNN-10-64$^{L=1}$ | 1.6e−6 | 4.1e−5 | 59.18 |
| TGV 3D | GNS-5-64 | 3.8e−4 | 8.3e−3 | 8.41 |
| | GNS-10-128 | 2.1e−4 | 5.8e−3 | 30.48 |
| | SEGNN-5-64$^{L=1}$ | 3.1e−4 | 7.7e−3 | 79.42 |
| | **SEGNN-10-64$^{\mathbf{L=1}}$** | 1.7e−4 | 5.2e−3 | 154.30 |
| RPF 3D | GNS-5-64 | 1.3e−6 | 5.2e−5 | 8.41 |
| | GNS-10-128 | 3.3e−7 | 1.9e−5 | 30.48 |
| | SEGNN-5-64$^{L=1}$ | 6.6e−7 | 3.1e−5 | 79.42 |
| | **SEGNN-10-64$^{\mathbf{L=1}}$** | 3.0e−7 | 1.8e−5 | 154.30 |
| | EGNN-5-128 | *unstable* | *unstable* | 250.67 |
| | PaiNN-5-128 | 1.8e−5 | 3.6e−4 | 43.98 |
| LDC 3D | GNS-5-64 | 1.7e−6 | 5.7e−5 | 8.63 |
| | **GNS-10-128** | 7.4e−7 | 4.0e−5 | 31.98 |
| | SEGNN-5-64$^{L=1}$ | 1.2e−6 | 4.8e−5 | 81.21 |
| | SEGNN-10-64$^{L=1}$ | 9.4e−7 | 4.4e−5 | 161.23 |

# I Training strategies

We explored the impact of the training tricks by training a 10-layer 128-dim latent space GNS model on the 2D RPF dataset and selecting the best model in terms of the 20-step validation MSE loss. We ran this setup in four configurations exploring the impact of random-walk noise (Noise) and push-forward (PF).

Table 8: Comparison of training with and without training strategies.

| Noise | PF | $MSE_5$ | $MSE_{20}$ | Sinkhorn | $MSE_{E_{kin}}$ |
|-------|-----|---------|-----------|----------|-----------------|
| yes | yes | 1.1e−7 | 3.3e−6 | 1.4e−7 | 1.7e−5 |
| no | yes | 1.0e−7 | 6.1e−6 | 1.1e−6 | 4.8e−5 |
| yes | no | 1.2e−7 | 3.8e−6 | 1.1e−7 | 2.8e−5 |
| no | no | 1.4e−7 | 8.3e−6 | 1.3e−6 | 2.4e−5 |

We observe that training without noise or push-forward results in significantly worse errors by all four measures. We also observe that combining the two strategies leads to the lowest $MSE_{20}$. Push-forward smoothens the training curves, while the additive noise prevents overfitting by adding fluctuations. It is, therefore, reasonable to expect that the training with up to 3 steps push-forward leads to the best performance on $MSE_5$, but it seems that the random noise is more important to recover the long-term behavior and the right particle distribution (lower Sinkhorn distance). Regarding the kinetic energy, it is hard to interpret the results, but it seems like a combination of both training strategies leads to the lowest error. Overall, including both training tricks seems to have a positive impact.

