# OpenReview forum: "LagrangeBench: A Lagrangian Fluid Mechanics Benchmarking Suite"
_NeurIPS.cc/2023/Track/Datasets_and_Benchmarks — NeurIPS 2023 Datasets and Benchmarks Poster_

### Official Review · Reviewer_6nz7 · 2023-07-14
**Review for 943 A Lagrangian Fluid Mechanics Benchmarking Suite: A well constructed and documented database for Lagrangian particle fluid problems with demonstrated utility**

**Rating:** 8
**Confidence:** 3

**Strengths:**

A. The overall overview of PDE methods and the difference between Lagrangian and Eulerian methods is well-constructed, well-organized, and provides useful references. This is especially true for readers (such as myself) who are not familiar with SPH and thus only have a surface-level understanding of Lagrangian methods.

B. Similarly, the overview of GNNs in section 3.2 is informative and well done.

C. The conclusion is also well-constructed in that it includes a discussion of key contributions as well as the identification of crucial and achievable avenues of future work.

Overall, it is my estimation that this is a well-designed study with significant contribution for the community.

**Additional Feedback:**

No additional feedback by the reviewer.

**Clarity:**

Overall, the paper is relatively clear and can be followed without extreme difficulty.

There are some improvements that I suggest in order to improve the overall message and readability of the manuscript:

A. On page 2, at the end of the "Eulerian Machine Learning" section, the last sentence has a lot compressed into it. I am understanding it as saying GNNs lag behind CNNs; also, GNNs are more suited for continuous space, and CNNs are more suited for Eulerian. Thus, Lagrangian ML methods have lagged behind their Eulerian counterparts. If possible, please restructure/split this last sentence into 2 sentences for clarity.

B. On page 2, in the "Lagrangian Machine Learning" section, please clarify what the phrase "hierarchical version" means in this context. Please provide reference or elaborate on meaning if possible. If it is known jargon, then please disregard. As someone who is more familiar with the fluids side of the jargon and not the ML side, I am not familiar with this term.

C. In the Lid-driven cavity paragraph of section 2.1, the last sentence discusses that the 3D section does not change the flow field significantly. Please provide additional justification as to why the 3D case is also included then? Is it to demonstrate the capabilities of the ML framework even for the larger 3D case? Whatever the reason, please elaborate.

D. In the Dam break paragraph of section 2.1, when discussion the small amount of physical velocity added to reduce artifacts, please provide some additional justification and elaboration (such as a reference if possible, of other studies that show this same behavior and/or treatment). I presume that the underlying cause appears largely numerical, but more details would provide clarity.

E. When discussing the size of datasets in the first few sentences of section 2.2, it would add clarity to include these sizes as a column to Table 1.

F. In the third paragraph of section 2.2, when saying that "JAX is faster on graph machine learning tasks at the time of writing this paper", please provide a reference for this statement and/or a figure or data to back this up, if possible. This could be added to the appendix, for example.

G. In section 3.1, it is stated that the nodes are connected based on an interaction radius of "~ 1.5 times". For my own knowledge, please clarify why it is only approximate. Is it because of finite size effects of the particles?

H. In section 3.3, when introducing the kinetic energy error, I see later on in the paper that this is a mean-square error of kinetic energy. Please specify this (i.e., how the kinetic energy error itself is computed).

I. In section 3.4, when saying that the GNS model is probably the fastest and most popular one, please add a reference to back up this claim, if possible.

J. In figure 2, I see that the x-axes ticks were spaced evenly even though the amount used is not uniformly spaced. Unless there was a reason why this was done (if so, please let me know), please adjust the x-axis ticks so that they reflect the numerical values of the "amount of available data" used.

K. In the second-to-last paragraph of section 3.5, it is stated at the end that both models (GNS and SEGNN) differ slightly. Please clarify what this means, do they differ slightly for the 2D vs. 3D? I did not understand the phrasing of the last sentence.

L. In the last paragraph of section 3.5, if possible, please provide some reasoning/rationale as to why EGNNs is seen to be slower than GNS, while PAINN is fast to run.

M. In the last paragraph of section 3.5, when stating that PaiNN has "three times the latent dimension", do you mean the dimensionality (number of dimensions?). Please clarify.

**Correctness:**

A. The SPH setup for the fluid mechanics, based on my eye as an Eulerian flow researcher (mostly at high Reynolds numbers), appears correct. I must rely upon the authors and other reviewers with more familiarity in the Lagrangian research field to ensure the validity and robustness of the setup.

B. Similarly, for the benchmarks, they appear to be soundly constructed as far as I could tell.

C. I have one question as clarification: because multiphase cases are present, shouldn't the surface tension force term be included in the right hand side of equation 2, such as in Zoller Adams Adami JCP 2023? If I am misunderstanding (these specifics of SPH equations is outside of my field) and/or if this term is included in the force field term F, please let me know. Otherwise, please correct this.

D. I also would like to point out one potential typo: In section 3.1, near the beginning the system particle positions P^t are defined with x's (and not p's) after the first entry. If this is correct, please disregard, as I misunderstood (and if you could please explain it to me, I would appreciate it). Otherwise, please correct this.




**Documentation:**

A. The overall database provided seems well-documented, with accompanying links, and the benchmarks seem sufficiently detailed. In section 2.2, it is stated that datasets comprising less than 8 GB are available at the given URL. Is it possible to make all datasets available? If not, please disregard, but this would be ideal for completeness.

B. It seems, as mentioned in section 2, that the code for the JAX API is not yet open-sourced and will be in the next few months. I don't believe this should count against the authors (for their candor and openness), but NEURIPs and the authors should work together to make sure this happens in the event of acceptance.

C. In section 3.5, the authors state that even though their JAX implementation is unsuccessful for EGNN and PaiNN, the implementation is included in the repository regardless to inform future research. I applaud this openness in documentation towards future research, thank you to the authors.


**Ethics:**

No ethical concerns are suspected by myself.

**Limitations:**

The authors provide justification and state their limitations (such as in the Decaying Taylor-Green Vortex paragraph of section 2.1, as an example). This is commendable, and so yes, I believe that overall the authors do state the limitations of their work when applicable.

I do not anticipate negative societal impacts resulting from this work, and none (as far as I could tell) were mentioned.

**Opportunities For Improvement:**

The opportunities for improvement that I suggest for this manuscript can be combined with the points of clarity, and so I list them there.

**Relation To Prior Work:**

The relation to prior datasets and data analysis is discussed near the end of page 2. However, the discussion is brief, and I request that these be elaborated upon. A key message of the current manuscript is how the current work is the first to create an establishing database (with accompanying framework) for Lagrangian fluid dynamics. Thus, it is important to demonstrate how the current work differs from what has been done. For the cited works in line 65 and 66, please describe how many datasets they contained, any distinguishing features, what is meant by not as diverse, which systems did they look at. If there are at all any other relatively close databases in the literature, please provide mention of them.

**Summary And Contributions:**

The paper sets the stage by pointing out that for the two main families of descriptions for numerical methods for PDES (Eulerian and Lagrangian),
machine learning PDE surrogate techniques for Lagrangian descriptions are relatively nacent when compared to techniques for Eulerian descriptions.

The author(s) then introduce the framework of their contributions that consist of 1) a set of 7 SPH-generated Lagrangian fluid dynamics datasets covering a range of physical behaviors and geometries, 2) a specifically-designed interface with training strategies, and 3) benchmarking results using the interface and results of established GNNs.

Specifically, the authors claim that this is, to their knowledge, the first formalized benchmarking for Lagrangian dynamics.

Benchmarking results are comprised of error measures and data scaling (inference time and memory usage).

---

> ### Author Response · Authors · 2023-08-19
>
> We thank the reviewer for the highly detailed feedback and constructive feedback on the manuscript. We try to answer all questions as detailed as possible.
>
> ### Section "Clarity"
> C. Actually, none of the presented datasets is a multi-phase case if we consider wall interactions as boundary conditions. The dam break represents a free surface problem.
>
> D. There was a typo. Thanks!
>
> ### Section "Correctness"
>
> A. **(lines 48-52)** We agree that the sentence is convoluted and rewrote it as: "We note that convolutional neural networks (CNNs) are well suited for problems in the Eulerian description, whereas graph neural networks (GNNs) are best suited to describe points in continuous space. Given the somewhat shifted uprise of research on GNNs with respect to CNNs, we conjecture that particle-based PDE surrogates are still in their early stages of development."
>
> B. "Interaction Networks [5] and their hierarchical version [43] ..." - we simply mean that ref. [43] extends the idea from [5] towards a graph U-Net hierarchy of short to long-range interactions by operating on graphs of different resolutions. Please refer to [43] for more details.
>
> C. **(181-184)** Yes, the case is intended to show that the nominally 3D flow recovers the 2D one. We include the 3D LDC dataset as a benchmark for methods that rely on 3D graphs, e.g. all codes that are based on the `e3nn` library.
>
> D.  "Artificial viscosity" is required to stabilize simulations (see [1] from the manuscript). With SPH there is not much difference between physical and artificial viscosity. We essentially add a very small amount of physical viscosity for stabilization.
>
> E. **(618 and Table 4)** Table 1 is already overcrowded. Nevertheless, we added the dataset sizes to Table 4 (extension of Table 1) in the appendix.
>
> F. **(207, Appendix E)** Thank you for the hint, we were also thinking of such a table in the appendix and now added it. We do get between 1.15x to 20x speedups.
>
> G. We reverse-engineered this radius of \~1.5 (to be more precise, we use 1.45) from the GNS paper and just stuck to it as the number of neighbors seems reasonable. The "\~" comes from rounding errors, i.e. our radius has two significant digits.
>
> H. **(310-311)** We compute the kinetic energy at each time step of 1) the rollout, and 2) the dataset. This gives us two arrays of scalars, between which we then compute the MSE, and we then average these values over all rollouts for the final results. We added this explanation to the paper.
>
> I. **(318)** "GNS model is probably the fastest and most popular one" - it is hard to find a reference for this claim, but the paper has been cited more than 700 times according to Google Scholar and it is in our experience more than 5x faster than SEGNN. The claim about the speed of GNS is based on the fact that GNS is one of the simplest GNN architectures possible, and by this, it does not have overheads common for example to hierarchical graph approaches.
>
> J. We set the x-ticks on purpose equidistantly to improve readability.
>
> K. **(350)** Based on the table in Appendix F, both models differ slightly on most datasets. What further investigations show is that it is not this much about 2D vs 3D, but rather that SEGNN underperforms on problems with wall boundaries. The reasons are still unclear.
>
> L. **(354-359)** The slow speed of EGNN is indeed a riddle to us. PaiNN is similar to GNS, but even after optimizing the EGNN implementation, there is still a large overhead closer to SEGNN. We noticed in profiling that a large portion of EGNN runtime is spent in computing spatial periodic displacements and shifts. More details in the revised manuscript.
>
> M. **(360-362)** In both the message and update blocks of PaiNN there is a 3x uplifting of the latent embeddings followed by a splitting into three different representations (related to scalar and vector features). This uplifting requires a large matrix multiplication and we suspect that this causes the large memory footprint of PaiNN.
>
> ### Section "Relation To Prior Work" **(68-75)**
> Out of the two papers on lines 65 and 66, the one by Li&Farimani has a dam break and a water fall dataset, the first of which is similar to our dam break. The Toshev et al. paper has a 3D TGV and 3D RPF. However, we are the first to propose the challenging dataset of predicting every 100th simulator step, on top of which we also introduce LDC. We will elaborate more on this in the paper.
>
> ### Section "Documentation"
> A. **(198)** All datasets together comprise 8GB and are all available under the provided link. We improved our formulation in the manuscript.
>
> B. We already open-sourced the code under [github.com/tumaer/lagrangebench](https://github.com/tumaer/lagrangebench), and documentation under [https://lagrangebench.readthedocs.io](https://lagrangebench.readthedocs.io).

---

> > ### Comment · Reviewer_6nz7 · 2023-08-20
> > **Response by Reviewer**
> >
> > Thank you to the authors for addressing each of my points. I will read over the revisions to the papers when they are made, but if each of the points are addressed in the manner that the authors have discussed, then I am satisfied.

---

> > > ### Comment · Reviewer_6nz7 · 2023-08-24
> > > **Repeat of my response to the top comment thread**
> > >
> > > As mentioned above, I've reviewed the changes made in response to my suggestions, and I am satisfied. Thank you.

---

### Official Review · Reviewer_YttU · 2023-07-20
**This a decent paper which applies Neural Networks to an under-researched field. The research is thoroughly done. However, the dataset could have been constructed with more relevant cases and the analysis of the benchmark results fall a bit short.**

**Rating:** 5
**Confidence:** 3
**Clarity:** The paper is very well written.

**Strengths:**

First of all, the application of NNs to fluid problems and more specifically to Lagrangian datasets is extremely interesting for the flow physics community. Even though the ML application might seems rudimentary compared to the state-of-the-art, there is a significant gap to be filled in order to apply successfully simple NNs to the problems considered in this paper.

The application to solve/predict partial differential equation (PDE) behavior is also very interesting and a fairly novel area of research that the authors tackle.

The research seems thoroughly conducted, and the claims are well supported by evidence. The paper is also extremely well written.


**Additional Feedback:**

N/A

**Correctness:**

The conclusion are sound and are based on tangible evidence. The dataset seems well constructed, albeit with the limitations that were previously mentioned. The benchmark results shown in the paper and appendix are reasonable but lack an in-depth investigation to explain some of the unexpected results or unstable behavior observed.

**Documentation:**

The dataset is well described. However, the appendix lacks a maintenance plan and its intended use. Sufficient references and details about the benchmark were provided for reproducibility.


**Limitations:**

The authors are upfront about the limitations on their dataset. However, I do think that a discussion on how to improve it is necessary (see the comment in the previous section).

It is evident that a lot of preliminary/non-ML work has been performed, leaving little time to the benchmarking and analysis of the NN performances. As a result, sections 3.5 and 4 fall a bit short in terms of explanation and discussion that are usually found in NEURIPS papers. Again, the authors are honest about the limitations of certain models and that some of them did not work as expected. Still, the lack of explanation on the performance of some NNs is disappointing.


**Opportunities For Improvement:**

I think that the datasets considered by the author are not of the highest interest for the research community in flow physics. Out of the 7 cases present in the dataset, only 1 features turbulence/unsteadiness of the flow. In addition, only the DAM case is really relevant for Lagrangian solver, since the other cases can be easily simulated in an Eulerian framework. In my opinion, the datasets could have been curated for the application of Lagrangian physics, especially considering that this dataset might be used by others to perform additional NN tests and development.

I do understand that this is a first step in the application of NNs to this kind of problem, and especially for the prediction of PDEs (which is an extremely challenging problem). However, there should at least be a discussion/plan, potentially in the appendix, of the additional cases that would be included in the datasets in the future.


**Relation To Prior Work:**

The use of NN for PDE prediction is well discussed both for Eulerian and Lagrangian solvers.

**Summary And Contributions:**

This paper focuses on the application of Neural Networks (NNs) for Lagrangian fluid mechanics problems. A relevant dataset is presented, along with an API for post-processing. In addition, a selection of well established NNs is tested to compare their performance and stability in the context of Lagrangian datasets.

---

> ### Author Response · Authors · 2023-08-19
>
> We thank the reviewer for the new perspectives and constructive feedback on the manuscript. We elaborate on the opportunities for improvement below.
>
> ### Dataset selection and future ideas **(lines 224-229)**
> With our dataset selection, we target flow problems which are well-established benchmarks in the flow physics community. At the same time, these cases offer diversity in terms of dominant mechanisms. More datasets with free surfaces and complex physics are desirable, and we currently work in this direction. We emphasize that our datasets provide high-quality quantitative reference solutions, unlike many earlier ones which are rather qualitative. Nevertheless, our framework allows everyone to plug in their datasets and train on them, as we have demonstrated with the datasets from the GNS paper in one of our notebooks: [github.com/tumaer/lagrangebench](https://github.com/tumaer/lagrangebench).
>
> Regarding future extensions of our datasets, we plan to add 1) a **multi-phase problem**, e.g. Rayleigh-Taylor Instability, and 2) a multi-phase flow **with surface tension**, e.g. drop deformation in shear flow. By multi-phase, we refer to fluids with different material properties. Introducing such complexity will require learning rich particle embeddings. Surface tension is a phenomenon that relates to how two different media interact with each other, and this would force the NN to learn complex interactions at low-dimensional manifolds.
>
> ### Discussion on benchmark results **(lines 258-263, 723-731)**
> We agree that the discussion on why EGNN and PaiNN didn't perform well could have been more exhaustive. However, we note that these two models have been designed to solve other problems by particle paradigms. We plan to benchmark a few more models like [1] and PointNet++. We did try to make EGNN and PaiNN work, as we explain in more detail in the response to reviewer **5Cwx**, but we didn't succeed by keeping the models somewhat close to their original design.
>
> ### Datasheets and maintenance plan
> We did spend a considerable amount of the manuscript describing the physics of the datasets, but we agree that a more standardized Datasheed in the appendix will be a good addition. Thank you for the idea and we would include it upon acceptance.
>
> As for the maintenance plan, in our understanding, we have already covered most of it in the manuscript. Upon acceptance, we would upload the datasets to Zenodo under the CC-BY license, and we do plan to extend them in the near future, which is rather easy to do with Zenodo.
>
> ---
> [1] - Prandtl et al., "Guaranteed Conservation of Momentum for Learning Particle-based Fluid Dynamics", 2022

---

### Official Review · Reviewer_5Cwx · 2023-07-21
**- The paper proposes a benchmark of datasets for Lagrangian Fluid Mechanics. I think the proposed work tries to fill the gap of providing a standard evaluation on newly developed machine learning models for Lagrangian dynamics and has the potential of doing so provided some minor-to-moderate improvements are conducted as detailed below.**

**Rating:** 7
**Confidence:** 3

**Strengths:**

- I think the paper identifies a gap of providing a well-diversified and well-studied engineering fluid mechanics datasets for Largrangian dynamics. Different datasets are built in order to cover a relatively large number of different 2D and 3D flows: stationary, decaying and free surface flows. Validation the used solver is well presented in supplementary materials. Four baseline models were tested against the proposed benchmark datasets. Three different error metrics are considered for models’ evaluation, including a physical error measuring the kinetic energy error. However, some improvements regarding the baseline models and error metrics can be applied as detailed below.

**Additional Feedback:**

Previous sections contain all feedback, comments, suggestions for improvement, and questions for the authors.

**Clarity:**

- The paper is well written and presented so that it conveys clear information and messages to be understood by the typical reader

**Correctness:**

- Most of the claims seem correct. However few points need some clarification as listed below:

- How baseline models hyperparameters were fixed? Different hyperparameters are tested based on tables 5, 6 and 7 in supplementary materials but some hyperparameters were always kept fixed (for instance MLP blocks, weight decay, LR decay steps, normalization, isotropic inputs for GNS etc). Why did authors keep all these hyperparameters fixed during their search? The same question applies for SEGNN (tables 6). For EGNN and PaiNN, only a singular hyperparameter configuration is considered for each of them. Authors should at least justify this configuration choice.

- I could not an explanation for PaiNN’s poor results (line 302). Authors should update this point if further clarifications were obtained in the meantime.

- In table 2’s title, it is mentioned that MSE_5 is taken as the validation loss. Could authors confirm that the 3 other error metrics reported in table 2 (MSE_{20}, Sinkhorn, MSE_{E{kin}}) are well evaluated on the test dataset? If not, authors must report some error metrics evaluated on unseen test dataset (and not only on validation dataset that was used to tune hyperparamters).

- Based on table 4 and the manuscript text describing the DAM break simulation (form line 161), it is unclear how the boundary condition is prescribed for this problem. Could authors specify it?


**Documentation:**

- Data and code were available through a lengthy series of files in the zip file uploaded as supplementary materials. A more comprehensive data organization would be needed if the manuscript is accepted (potentially with a Zenodo upload as stated in line 177).

**Ethics:**

- No, I could not suspect any ethical concerns with the submission that would warrant further discussion or review.

**Limitations:**

- I could not identify potential negative societal impact.

**Opportunities For Improvement:**

- TGV demonstrates the onset of turbulence but are fully turbulent flows considered among the seven built datasets?

- ML-based models that preserve conservation laws are of great interest in the machine learning-based computations community. Would it be possible to consider some baseline model(s) that preserve known physical laws?

- Related to the previous point, authors should consider error metrics for physical laws conservation (conservation of energy or momentum equation). Even for standard ML baseline models that are not designed to conserve these laws, it is very interesting to compare their behavior with respect to this aspect.

- All baseline models are variants or improved versions of GNNs. Could authors consider other ML models that are not GNN-based?


**Relation To Prior Work:**

- Previous proposed benchmark datasets for Lagrangian dynamics are well referenced and the proposed benchmark is compared against these references.

**Summary And Contributions:**

- The paper proposes a benchmark of datasets for Lagrangian Fluid Mechanics. There is a total of 7 datasets that are built based on 4 different physical problems prescribed by PDEs that are 2 or 3 dimensional in space. These problems span different scenarios including: stationary flows, decaying flows and flows with free surface. Four different baseline models are tested on the proposed datasets. All these models are variants or improved versions of Graph Neural Networks. An additional contribution of the proposed work is the extension of PaiNN to vectorial inputs. However, the relatively poor PaiNN results on the benchmark dataset compared to other less expensive GNN-based models weakens the value of this contribution. Three different error metrics are considered for models’ evaluation, including a physical error measuring the kinetic energy error.

---

> ### Author Response · Authors · 2023-08-19
>
> We thank the reviewer for the insightful comments and constructive feedback on the manuscript. We elaborate on the opportunities for improvement below.
>
> ### Turbulence modeling
> Our datasets are based on fully resolving the smallest scales and directly solving the NSE. For practical reasons, we currently limit the maximum particle number to about 10000, which precludes the simulation of resolved turbulence. We point out, however, that non-turbulent flows may be more informative for training surrogate Navier-Stokes dynamics as low-frequency statistics of e.g. isotropic turbulent flows actually are quite similar.
>
> ### Conservation laws
> We solve the weakly-compressible NSE and use total kinetic energy as a validation metric. We do compare results in terms of this conserved quantity. We agree that the local conservation of quantities like linear and angular momentum are of great interest, and they have indeed shown huge potential for learning Lagrangian fluid dynamics [1].
>
> ### Ideas on more baselines **(258-263)**
> Our framework is general enough to allow for any type of model independent of whether it is GNN-based. The momentum-preserving model from [1] is being integrated into LagrangeBench. The same applies to PointNet++.
>
> We also considered (graph) transformers but did not further pursue this direction as they would not learn local interactions and therefore would not be scalable to arbitrarily large systems. We want to stress that GNNs are one of the few paradigms that can handle point clouds and incorporate information from adjacent particles.
>
> ### Clarification on included baseline configurations **(342-343, 709-731)**
> Training the baselines takes several days, especially the larger instances of SEGNN. In order to reduce the amount of training runs we prioritized structural/model-related values over optimization hyperparameters like LR decay rate. We did investigate in-depth non-isotropic input normalization and found it advantageous since normalizing the input dimensions independently leads to a more favorable data distribution. However, because independent rescaling of vector dimensions via normalization breaks equivariance, all equivariant models use isotropic normalization by default.
>
> **EGNN and PaiNN**
> Both EGNN and PaiNN were evaluated even beyond hyperparameters variations:
> - For EGNN, we found that going beyond 5 message passing steps resulted in unstable gradients. The poor stability of EGNN on larger systems was already documented in [2]. We tried two conceptually different implementations: 1) original N-body formulation, and 2) a layer directly working with velocities without updating positions. But we found that only the first approach behaved well.
> - As for PaiNN, we tried different setups with/without cutoff functions and radial embeddings, including different features in the nodes and with/without our change, which allows for general vector inputs. We observed that a (cosine) cutoff of 1.5 times the average inter-particle distance, together with a learnable Gaussian RBF produced the most reasonable results.
>
> The two EGNN and PaiNN configurations in the Appendix of our paper are, unfortunately, still the best ones we have. In the future, we will be working on developing better models, but it seems that these two baselines do not work out of the box, mainly because they were designed to tackle different problems/systems.
>
> ### Details on Table 2 values **(Table 2)**
> We already improved our formulation in the manuscript. The training procedure within our API involves 1) checkpointing the current model weights at every evaluation on the validation dataset and 2) checkpointing the weights that performed best to date on the validation loss. It is this latter best checkpoint that we select based on the 5-step validation MSE. Once training has finished, we evaluate the best checkpoint on the full test set and these are the results we report in Table 2.
>
> ### Solid wall boundary implementation **(143-149)**
> Both lid-driven cavity and dam break have solid wall boundaries, implemented using the established generalized wall boundary conditions approach [3]. This approach relies on representing the walls with "dummy" particles and then 1) enforcing a no-slip boundary condition at the wall surface by assigning to the wall particles the opposite velocity of that of the closest fluid particles, and 2) enforcing impermeability by assigning the same pressure to the wall particles as the pressure of the closest fluid. This approach requires using multiple layers of wall particles, but we make the ML task even more difficult by leaving only the innermost wall layer in the dataset.
>
> ---
> [1] - Prandtl et al., "Guaranteed Conservation of Momentum for Learning Particle-based Fluid Dynamics", 2022
>
> [2] - Brandstetter et al., "Geometric and Physical Quantities Improve E(3) Equivariant Message Passing", 2022
>
> [3] - Adami et al., "A generalized wall boundary condition for smoothed particle hydrodynamics", 2012

---

> > ### Comment · Reviewer_5Cwx · 2023-08-20
> >
> > Thank you for these clarification. I think some of the answers provided above are quite relevant and could be included in the paper if accepted (since you will have one more page in terms of spacing, or in supplementary materials if needed).

---

### Official Review · Reviewer_uzym · 2023-08-01
**A review of a LagrangeBench**

**Rating:** 6
**Confidence:** 2
**Clarity:** I have no specific concerns regarding…

**Strengths:**

The objective of the paper is clear, and it seems to occupy the specific niche in scientific computing which was not fully covered by previous works. A set of GNN-based solvers were implemented and evaluated on the provided benchmark data.

**Additional Feedback:**

No special feedback

**Correctness:**

The claims made seem to be correct, but I need more clarifications regarding efficient JAX-based API (second point in the contributions). I didn’t find details on this API neither in the paper nor in the code.

**Documentation:**

The sufficient details on the dataset are provided. The baseline methods are implemented.

**Ethics:**

No concerns

**Limitations:**

The limitations and potential negative societal impact is not covered in the paper, but I don’t consider it as a drawback. The benchmark deals with a specific scientific problem (modelling NS dynamics), and creating a new benchmark for such a sphere is always a step towards better reliability and robustness of newly proposed methods.

**Opportunities For Improvement:**

To be honest, I am not a specialist in the field of knowledge the authors address (solving NSEs with mesh/particle - based methods), that is why it is somewhat difficult for me to estimate the real contribution to community, soundness and efforts to conduct all the stuff required to generate the data. As I understand, the authors just take several well-known physical systems, governed by concrete Navier-Stokes equations and simulate the data using the well-known SPH method. That is it (omitting JAX-based reimplementations of several known GNNs). I do not see significant novelty in the proposed work: the paper proposes just a standardized set of data to train and evaluate the models based on Lagrangian schemes. Importantly, there is no mystery in the data itself: I guess, previous researches generate similar data similarly.

More specific points are given below:
- The paper seems to be not completed: There is not any text in section D (appendix). This appendix should describe the details on Neighbors search implementations via JAX.
- There are no details on the Smoothed Particle hydrodynamics method: how it works, why can it be considered as a good approximation of governing NS equations? This aspect seems to be very important for the benchmark construction.
- The related literature regarding baseline methods is not properly covered: the authors consider several GNNs, but it is not clear why they choose these specific models. More extensive analysis of existing solvers which could be used for the problem the authors deal with should be done.
- In the contributions listing, authors state that they provide “Efficient JAX-based API  with various recent training strategies including additive random walk-type input noise and the pushforward trick … “. I didn’t find details on this API in the remaining paper, and I am not sure if it is implemented in the provided code.

**Relation To Prior Work:**

There is discussion about existing works, but I am not sure if it is complete and covers all necessary researches

**Summary And Contributions:**

The authors consider several well-known physical setups governed by Navier-Stokes equations. They simulate the corresponding PDEs using smoothed particle hydrodynamics and utilize the obtained data as the benchmark datasets, which can be used for evaluating machine learning methods designed to solve such physical problems by means of Lagrangian (particle-based) scheme. The authors specifically focus on GNNs and consider several popular GNN models as baselines. Apart from the dataset suite itself, the authors plan to open-source their own JAX-based implementations of GNN solvers.

---

> ### Author Response · Authors · 2023-08-19
>
> We thank the reviewer for the new perspectives and constructive feedback on the manuscript. We elaborate on the opportunities for improvement below.
>
> ### Novelty
> Indeed, we do (1) take known Navier-Stokes systems, (2) simulate them with SPH to generate datasets, and (3) implement known GNNs in a JAX-based API. However, there are major differences between what we propose and what exists in the ML literature.
> - Regarding (1), **we chose systems that are established benchmarks in fluid mechanics research, but go beyond systems typically considered in the ML community** in terms of physical complexity. This applies in particular to treatment with particles-based approximations.
> - Regarding (2), we stress that **our datasets are purely Lagrangian**, in contrast to many other SPH implementations like [1], which are approximately Lagrangian, i.e. their particles do not follow the physical velocity field in order to manipulate the particle distributions. It is necessary to be Lagrangian if we want to learn surrogate dynamics based on physically meaningful loss definitions. To the best of our knowledge, other Lagrangian ML papers do not even discuss such aspects.
> - Regarding (3), indeed, most of our models are JAX reimplementations except for the adapted PaiNN. But we benchmark all these models on complex fluid datasets and provide a highly flexible and efficient API with recent training strategies.
> - (4) - introducing physically interpretable losses like Sinkhorn distance and kinetic energy is another major contribution of our benchmarking suite, as the **considered physical systems are highly chaotic, learning based on global/statistical properties, in this case, is more appropriate** than trying to predict the individual realizations of particle positions.
>
> ### Neighbors' search details **(Appendix D)**
> We have now completed Appendix D. The mistake was caused by accidentally uploading a preliminary version. We also wrote a README (`neighbors_search/README.md`) on the neighbors' search backends in our now public repository: [github.com/tumaer/lagrangebench](https://github.com/tumaer/lagrangebench).
>
> ### Why SPH? **(lines 101-107)**
> SPH is a very well-established Lagrangian approximation method of the Navier-Stokes equations. It has been introduced in the 70s and has become a standard method for numerical fluid mechanics. The core idea of SPH is to discretize a domain with fluid particles and define the properties of the fluid at some particular locations through a truncated radial kernel interpolation over neighboring particles. By rewriting the NSE in terms of kernel interpolations of the fluid properties (i.e. velocity, pressure, density), we arrive at a system of ODEs for the particle accelerations. By integrating twice we update the velocities and positions of the particles. For more details on the particular implementation, we refer to [2].
>
> ### Baselines **(lines 258-263)**
> First, we want to stress that GNNs are one of the few paradigms that can handle point clouds and incorporate relational information between adjacent nodes. One potential alternative would be transformer models, but they would not learn local interactions and are therefore not scalable to arbitrarily large systems. To the best of our knowledge, the only real alternative to GNNs is continuous convolutions [3], and we plan on integrating such models into our codebase in the near future. Our API is not limited to GNNs and adding further models is rather straightforward.
>
> Regarding alternative GNN models, we believe that we have already included the two most relevant models GNS and SEGNN, and we are working on adding simpler models like PointNet++ and Graph U-Nets.
>
> ### Documentation
> We now added **more extensive documentation and three tutorial notebooks** to our codebase [github.com/tumaer/lagrangebench](https://github.com/tumaer/lagrangebench). Also, we created a proper documentation website ([https://lagrangebench.readthedocs.io](https://lagrangebench.readthedocs.io)) and a publicly available PyPI instance of the code ([https://pypi.org/project/lagrangebench](https://pypi.org/project/lagrangebench)).
>
> ---
> [1] - Zhang et al., "A generalized transport-velocity formulation for smoothed particle hydrodynamics", 2017
>
> [2] - Adami et al., "A generalized wall boundary condition for smoothed particle hydrodynamics", 2012
>
> [3] - Ummenhofer et al., "Lagrangian Fluid Simulation with Continuous Convolutions", ICLR 2020

---

> > ### Comment · Reviewer_uzym · 2023-08-26
> > **Concluding answer to the authors**
> >
> > Dear authors. Thank you for your answer. While the real contribution and groundbreaking of your research are still somewhat beyond my expertise but seem to be limited, all of my concerns are more or less satisfied. Additionally, I really appreciate that the code is open-sourced and covered by (seems to) extensive documentation. That is why I consider raising my score by one. FYI: I am looking forward to adding new baselines to the benchmark. It will contribute to the work.

---

### Official Review · Reviewer_Dee9 · 2023-08-02
**LagrangeBench: A Lagrangian Fluid Mechanics Benchmarking Suite**

**Rating:** 6
**Confidence:** 4
**Correctness:** see "Opportunities For Improvement"
**Clarity:** Yes

**Strengths:**

- This paper provides seven new fluid dynamics datasets to evaluate machine learning models for the Lagrangian-based fluid simulation. These datasets include 2D and 3D Taylor-Green vortex (TGV), 2D and 3D reverse Poiseuille flow (RPF), 2D and 3D lid-driven cavity (LDC), and 2D dam break (DAM). These datasets cover different dynamics, such as the onset of turbulence, spatially dependent external force field, static and moving wall boundaries, and free surface.
- This benchmark provides JAX-based API with various training strategies including noising injection and pushforward trick.
- This paper provides JAX implementation and baseline results of established GNNs, including GNS, EGNN, SEGNN, and an adapted PaiNN model.

**Additional Feedback:**

see "Opportunities For Improvement"

**Documentation:**

Not enough documentation and code for data generation

**Limitations:**

The limitations of this paper are well discussed.

**Opportunities For Improvement:**

- I recommend authors build a public github repo and attach the link to the paper. In addition, it is better to provide a jupyter notebook to demonstrate how to load data, process data, train the model, and evaluate the performance, as this will be very useful for users to quickly use the dataset.
- This paper chooses some equivariant models as baselines. It is better to demonstrate whether these datasets exhibit equivariance property and if this is the reason that equivariant models perform better than GNS.
- This paper mentioned several training tricks, such as noise injection and pushforward. A very interesting experiment is to compare the effect of these training tricks on the long rollout performance of each model.

**Relation To Prior Work:**

Yes

**Summary And Contributions:**

This paper proposes LagrangeBench, the first benchmarking suite for Lagrangian particle problems. LagrangeBench includes seven new fluid dynamics datasets (four in 2D, three in 3D). These datasets are generated with the Smoothed Particle Hydrodynamics (SPH) method and each of which includes different physics like solid wall interactions or free surface. This benchmark provides efficient JAX-based API with various recent training strategies and neighbors search routine. This paper also benchmarks several baseline methods such as GNS and SEGNN.

---

> ### Author Response · Authors · 2023-08-19
>
> We thank the reviewer for the insightful and constructive feedback on the manuscript. We elaborate on the opportunities for improvement below.
>
> ### Public release of the code
> We initially submitted our work double-blind, but it seems that uploading the code to GitHub will make the review process easier. That is why we now made the code publicly available under the following URL: [github.com/tumaer/lagrangebench](https://github.com/tumaer/lagrangebench). And this is the link to our documentation website: [https://lagrangebench.readthedocs.io](https://lagrangebench.readthedocs.io).
>
> In addition, we already included **more extensive documentation and three tutorial notebooks**, going over all necessary LagrangeBench components: data access, model setup, training configuration and application of training tricks, and finally evaluation/rollout generation.
>
> ### Equivariance of data
> The dynamics describing our datasets are fully determined by the Navier-Stokes-Equations (NSE). Knowing that the NSE dynamics are equivariant with respect to isometries of  Euclidean space (rotations, translations, and reflections), the question is whether these symmetries are broken in our datasets. To the best of our knowledge, the answer to this question is no. First of all, the ground truth data simulator is based on the Smoothed Particle Hydrodynamics (SPH) method and we know that it preserves equivariance. Secondly, another way to break symmetry could be non-isotropic material properties where "approximate equivariance" has demonstrated promising results [1], but in our case, the only material property used for data generation is viscosity, which is constant and isotropic over the whole domain. Lastly, the equivariance regarding external forces is also preserved as we pass such directional node features to SEGNN as vectors.
>
> ### Training tricks and long-term behavior **(lines 332-334 and Appendix H)**
> We explored the impact of the training tricks by training a 10-layer 64-dim latent space GNS model and selecting the best model in terms of the 20-step MSE validation loss (in contrast to Table 2, where we select based on the 5-step MSE). We ran this setup in four configurations exploring the impact of noise and push-forward (PF). We added the results to the appendix and summarize them also here:
>
> | noise | PF  | MSE_20               | Sinkhorn             | MSE_e_kin            |
> | ----- | --- | -------------------- | -------------------- | -------------------- |
> | yes   | yes | $6.25e-6\pm1.3e-6$ | $3.67e-7\pm1.3e-7$ | $1.63e-5\pm2.0e-5$ |
> | no    | yes | $7.33e-6\pm1.5e-6$ | $3.85e-7\pm1.2e-7$ | $2.09e-5\pm2.5e-5$ |
> | yes   | no  | $6.31e-6\pm1.3e-6$ | $3.90e-7\pm1.2e-7$ | $2.45e-5\pm3.2e-5$ |
> | no    | no  | $8.54e-6\pm1.7e-6$ | $5.52e-7\pm2.0e-7$ | $1.85e-5\pm2.3e-5$ |
>
> We observe that training without noise or push-forward results in significantly worse 20-step MSE and Sinkhorn distances. We also observed that combining the two strategies does not cause problems in training. Push-forward smoothens the training curves, while the additive noise prevents overfitting by adding functions. The difference between only noise perturbations or only push-forward is rather marginal, but it seems like the noise is more crucial than push-forward for the position errors. Regarding the kinetic energy, it is hard to say how it is influenced as the uncertainty of the results is larger than the difference between methods. Overall, including the training tricks seems to have a positive impact.
>
> ---
> [1] - Approximately Equivariant Networks for Imperfectly Symmetric Dynamics, Wang et al., ICML 2022

---

> > ### Comment · Reviewer_Dee9 · 2023-08-24
> >
> > My concerns have been well addressed, so I will increase my score.

---

### Author Response · Authors · 2023-08-22
**General Response**

We want to express our sincere gratitude to the reviewers for their invaluable and insightful comments, which significantly contributed to the improvement of the manuscript. We have responded to each review separately, and summarize the central points of the reviews here.

We just uploaded our **revised manuscript** with all changes based on the reviews. Also, we edited all our previous responses by adding the relevant line numbers from our revised manuscript.

### Public codebase

1. Our code is now publicly available on GitHub: [github.com/tumaer/lagrangebench](https://github.com/tumaer/lagrangebench).
2. We also added a ReadTheDocs documentation: [https://lagrangebench.readthedocs.io](https://lagrangebench.readthedocs.io).
3. And finally, our code can be installed as a standalone library (instead of having to work within a clone of LagrangeBench), and we published it on PiPI: [https://pypi.org/project/lagrangebench](https://pypi.org/project/lagrangebench).

In addition, we also included **more extensive documentation and three tutorial notebooks**, going over all necessary LagrangeBench components: data access, model setup, training configuration and application of training tricks, and finally evaluation/rollout generation.

### Improvements to the revised paper
- **Ablation study on the impact of training tricks**. We run an experiment on the performance of the GNS-10-64 model on the 2D RPF when trained with/without random-walk noise and with/without push-forward. **(Dee9)**
- **Completed Appendix D** **(uzym)**
- **Speed comparison JAX vs PyTorch in Appendix** **(6nz7)**
- **Relation to existing datasets** **(6nz7)**
- **We have revised parts of the manuscript to improve clarity**:
    - On baseline selection. **(uzym, 5Cwx)**
    - On baseline configuration/results **(5Cwx, YttU)**
    - Caption of Table 2 **(5Cwx)**
    - Wall boundary conditions implementation **(5Cwx)**
    - Plan on future datasets **(YttU)**
    - Dataset sizes in Table 4 **(6nz7)**
    - Computation of kinetic energy **(6nz7)**
    - Many small changes based on the feedback from reviewer **6nz7**

---

> ### Comment · Reviewer_6nz7 · 2023-08-22
> **Response by reviewer to updated manuscript**
>
> I've reviewed the changes made in response to my suggestions, and I am satisfied. Thank you.

---

### Decision · Program_Chairs · 2023-09-22

**Decision:**

Accept (Poster)

**Comment:**

This work proposed a benchmark for ML modeling of PDE and fluid mechanics problems, which is a new and emerging area. The benchmark work is solid with positive supports from expert reviewers. Thus an accept is recommended.